# Beyond Correctness: Benchmarking Multi-dimensional Code Generation for Large Language Models

## Abstract

In recent years, researchers have proposed numerous benchmarks to evaluate the impressive coding capabilities of large language models (LLMs). However, current benchmarks primarily assess the accuracy of LLM-generated code, while neglecting other critical dimensions that also significantly impact code quality in real-world development. Moreover, relying exclusively on correctness as the guiding metric renders LLMs susceptible to data contamination. Therefore, this paper proposes the **RACE** benchmark, which comprehensively evaluates the quality of code generated by LLMs across 4 dimensions: **R**eadability, m**A**intainability, **C**orrectness, and **E**fficiency. Specifically, considering the *demand-dependent* nature of dimensions beyond correctness, we design various types of user requirements for each dimension to assess the model's ability to generate correct code that also meets user demands. We analyze 28 representative LLMs based on RACE and find that: 1) current correctness-centric benchmarks fail to capture the multifaceted requirements of code in real-world scenarios, while RACE provides a comprehensive evaluation that reveals the defects of LLMs across multiple dimensions; 2) the RACE benchmark serves as an effective tool for resisting the risk of data contamination; 3) even the most advanced code LLMs still encounter significant challenges in customized requirements involving complex instructions; 4) most LLMs exhibit an inherent preference for specific coding style. These findings highlight the need for a multidimensional evaluation of code LLMs, emphasizing metrics beyond correctness for real-world applications. Future efforts should aim to develop novel learning algorithms to enhance code generation under varied constraints and improve coverage and usability for diverse user needs[1].

## 1 Introduction

The impressive coding capabilities demonstrated by Large Language Models (LLMs) are reshaping the landscape of software development (Zheng et al., 2023c;b; Fan et al., 2023), attracting significant attention from researchers. To accurately measure and compare the coding capabilities of various LLMs, numerous benchmarks have been proposed to evaluate the code generation (Chen et al., 2021; Austin et al., 2021; Hendrycks et al., 2021), completion (Gong et al., 2024), and execution (Jain et al., 2024a) abilities of LLMs.

However, current benchmarks primarily focus on evaluating the correctness of LLM-generated code, while neglecting other critical dimensions that also significantly impact code quality in real-world development scenarios. For example, Börstler et al. (2023) investigate various aspects of code quality and find that code readability is the most decisive property for high-quality code (Dantas et al., 2023; Oliveira et al., 2020). Additionally, code maintainability is crucial for ensuring the software remains adaptable and easy to update, ultimately reducing long-term costs and technical debt (Hegedus, 2013). Code efficiency is essential for optimizing performance, reducing resource consumption, and ensuring scalability in software applications (Curtis et al., 2022; Börstler et al., 2023). As shown in Figure 1, current benchmarks lack evaluation on these critical dimensions that impact code qual-

---

[1]We provide our benchmark and source code at this anonymous repository `https://anonymous.4open.science/r/RACE-4AF1`.

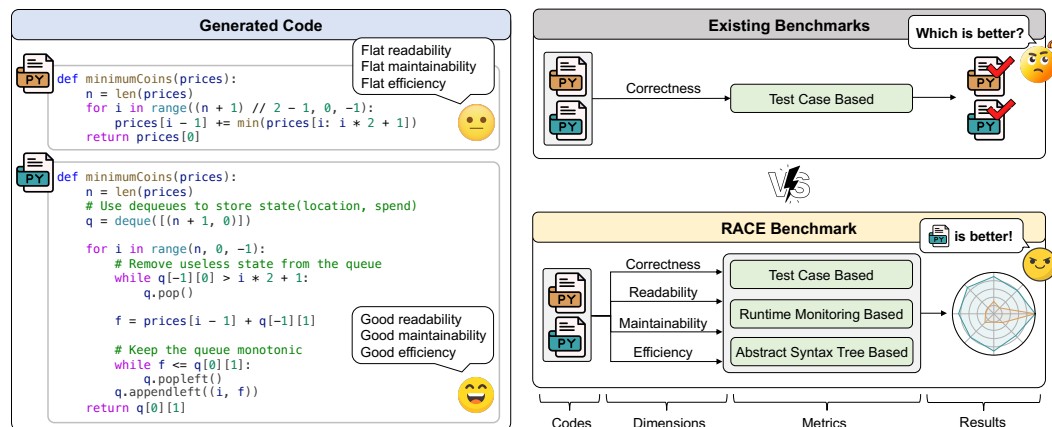

Figure 1: Current benchmarks perform single-dimension evaluations and mostly focus only on code correctness (upper right); our proposed RACE benchmark performs multi-dimensional code evaluations to identify truly high-quality code beyond correctness (lower right).

ity, making it challenging to distinguish genuinely high-quality code from merely correct code. Such deficiency in evaluation could lead to incomplete assessments of the coding capabilities of different LLMs in real-world development scenarios. Furthermore, if these correctness-based benchmarks serve as guiding indicators and correctness alone becomes the sole criterion for driving LLM development, these models might end up memorizing the exact solutions from the training data instead of understanding the underlying principles or patterns. This overfitting implies the model may reproduce code that is highly similar to the training data during inference, leading to data leakage. Consequently, this singular focus on correctness can render LLMs susceptible to data contamination, which has been proven to be quite prevalent due to the exponential scaling of pre-training data (Riddell et al., 2024; Cao et al., 2024). Therefore, there is an urgent need for a multidimensional code evaluation benchmark that transcends correctness, addressing the gap between LLM-generated code and real-world scenarios, and steering code LLMs towards comprehensive development.

To this end, we propose the **RACE** benchmark, designed to comprehensively evaluate the code generated by LLMs across multiple dimensions including **R**eadability, m**A**intainability, **C**orrectness, and **E**fficiency. However, it is not trivial to develop a reliable multi-dimensional benchmark for code generation. The first challenge is to design a quantifiable evaluation framework with corresponding metrics for each dimension. Unlike correctness, other dimensions are typically difficult to quantify with a single metric (e.g., accuracy). To address this, we refer to the definition of readability, maintainability, and efficiency in various quality models (Curtis et al., 2022; Nistala et al., 2019; Sadeghzadeh Hemayati & Rashidi, 2017), and summarize multiple representative factors for each dimension of code quality. Furthermore, dimensions beyond correctness cannot directly use the pass rate of test cases as the performance metric. Therefore, we develop specific evaluation metrics for each factor within these dimensions, which can be objectively and automatically calculated based on static analysis and runtime monitoring methods. As illustrated in Figure 1 and 2, by integrating performance across multiple factors, we can comprehensively assess the quality of LLM-generated code in each dimension. Another more critical challenge is that dimensions other than correctness are **_demand-dependent_**. This means a fixed and uniform standard cannot be used to assess what constitutes better code. Instead, different application scenarios could have varying requirements for code generation. For instance, various projects require unique coding styles and interface standards for adaptability and scalability. Additionally, balancing time and space efficiency based on hardware conditions ensures code operates efficiently. Therefore, a genuinely practical model should generate correct, customizable code that meets multiple dimensional requirements. To achieve this, we design various demands for each factor and incorporate them into the task descriptions, requiring the model to generate code that is both correct and meets the specified requirements. For example, we design multiple instructions that direct the model to generate multiple versions of code, each optimized differently for time and space efficiency. By incorporating the aforementioned quantifiable evaluation framework, we can accurately and efficiently quantify the extent to which LLM-generated code fulfills the corresponding customized requirements across each dimension.

Based on the RACE benchmark, we conduct a comprehensive evaluation and systematic analysis of 28 LLMs across various scales, encompassing the most advanced open-source (e,g, Qwen2.5-

72B (Yang et al., 2024) and DeepSeek-V2.5 (Zhu et al., 2024)) and closed-source models (e.g., GPT-4o (OpenAI, 2024a), o1-mini (OpenAI, 2024b) and Claude-3.5-Sonnet) in terms of coding capabilities. Our findings reveal that **using correctness as the only guiding indicator is insufficient for code benchmarks to steer code LLMs towards comprehensive advancement**. 1) Current benchmarks fail to capture the multifaceted requirements of code in real-world scenarios. Therefore, current code LLMs, developed with a primary focus on correctness, exhibit significant room for improvement in other critical dimensions, including code readability, maintainability, and efficiency (§4.2). 2) We present concrete evidence that current code benchmarks are susceptible to data contamination, compromising the fairness and reliability of the evaluation conclusions. In contrast, a contaminated model may simply reproduce memorized code without the ability to generate diverse solutions that meet various user requirements. Therefore, the RACE benchmark can robustly provide stable assessment results even under data contamination settings (§4.3). Moreover, **a deeper analysis based on RACE reveals notable deficiencies in current code LLMs**: 3) Even the most advanced code LLMs severely struggle to understand and follow complex instructions that include several customization requirements, with performance deteriorating significantly as the number of requirements increases (§4.4). 4) Most LLMs exhibit an inherent preference for specific coding styles, making it difficult for them to follow user instructions that are inconsistent with their preference (§4.5). The findings above highlight the importance of a multidimensional evaluation of code LLMs, while also revealing the necessity for metrics that extend beyond correctness to guide the development in real-world scenarios. In the future, code LLMs will require the design of novel learning algorithms to acquire the ability to generate high-quality code subject to multidimensional constraints, as well as to enhance their coverage and usability concerning diverse user requirements.

The main contributions of this paper can be summarized as follows:

- We propose a novel multi-dimensional evaluation framework for code generation.

- Based on the framework, we construct a comprehensive benchmark named RACE, featuring data construction, customized requirement instructions, and specific evaluation metrics.

- We evaluate and analyze 28 LLMs on the RACE benchmark, and obtain valuable conclusions that reveal the limitations of current benchmarks and models.

## 2 RELATED WORK

### 2.1 CODE LLMS

The outstanding code generation capabilities exhibited by LLMs have attracted considerable attention from researchers (Wang et al., 2021; Li et al., 2022; Fried et al., 2022; Xu et al., 2022; Roziere et al., 2023; Zheng et al., 2023a). Some representative code LLMs, such as CodeX (Chen et al., 2021), CodeGen (Nijkamp et al., 2022), and AlphaCode (Li et al., 2022), have achieved notable performance in code generation, program repair, and code translation. Currently, research on LLMs for code primarily focuses on data and pretraining methods. For training data collection, Wizard-Coder (Luo et al., 2024) introduces code instruction-following training constructed by Evol-Instruct to enhance the capabilities of code LLMs. For pretraining methods, StarCoder (Li et al., 2023a) and DeepSeek-Coder (Guo et al., 2024) incorporate fill-in-the-middle training task to enhance the model's capability to handle various structural arrangements in code. With the rapid advancement of code LLM capabilities, there is an increasing demand for reliable and comprehensive code evaluation benchmarks.

### 2.2 CODING BENCHMARK FOR LLMS

The existing benchmarks for LLM-based code (Ni et al., 2023), such as HumanEval (Chen et al., 2021), APPS (Hendrycks et al., 2021), MBPP (Austin et al., 2021), CodeContests (Li et al., 2022), and DS-1000 (Lai et al., 2023), focusing on the correctness of generated code in scenarios such as code exercises, data science, and competitions (Yan et al., 2023; Li et al., 2023b; Shinn et al., 2024). However, these efforts only focus on the correctness of the generated code, using the pass rate of test cases as the sole evaluation metric. Meanwhile, there has been a recent trend in considering other dimensions (Li et al., 2024; Jain et al., 2024b; Tian et al., 2024); for example, Huang et al. (2024) evaluate the efficiency of the generated code, while Dillmann et al. (2024) bridge the con-

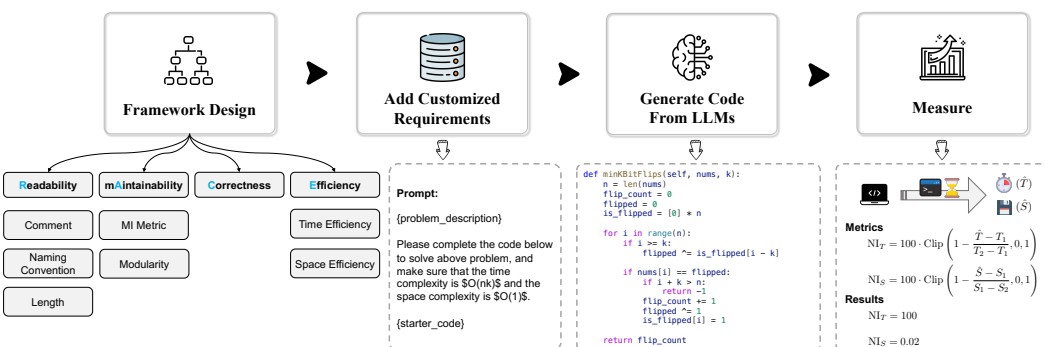

Figure 2: The overall evaluation pipeline in RACE benchmark.

nection between cross-entropy and logical lines of code. Nevertheless, these studies neither account for the demand-dependent nature of these dimensions nor systematically evaluate the LLM's code capabilities across multiple dimensions.

## 3 RACE BENCHMARK CONSTRUCTION

The philosophy of our framework design comes from the demands for code quality in software engineering (Börstler et al., 2023). Firstly, we summarize multiple representative factors for each dimension based on their respective quality definitions (Curtis et al., 2022; Nistala et al., 2019; Sadeghzadeh Hemayati & Rashidi, 2017). Secondly, we design several reasonable customized requirements for each factor and integrate them into task descriptions, requiring the model to generate code that is both correct and meets these requirements. Information on the detailed evaluation data is presented in Table 1. Finally, leveraging static analysis and runtime monitoring techniques, we develop evaluation metrics tailored to each factor to achieve accurate and efficient evaluation. The specific designs of each instruction refer to Appendix B.2.

### 3.1 CORRECTNESS

Correctness is the core and foundation for evaluating whether the functionality of code generated by models meets expectations. Therefore, to comprehensively assess the capability of LLMs in generating functionally correct code across various task scenarios, we select 4 datasets with different distributions: HumanEval+ and MBPP+ (Liu et al., 2024) for code exercise problems, ClassEval (Du et al., 2023) for class-level code generation, and LeetCode (Guo et al., 2024) for coding competition problems. To mitigate bias from extraneous information in the original dataset affecting the customized requirements, we exclude such information from the datasets. We use the macro accuracy across 4 datasets as the metric for correctness.

Furthermore, to investigate the impact of adding customized demands on code correctness, we also calculate the accuracy of the generated code when instructions with customized requirements are provided.

Table 1: The sources and number of evaluation cases for each factor in the RACE benchmark.

| Factors | Data Source | # Cases |
|---|---|---|
| **Correctness** | | |
| Correctness | HumanEval+, MBPP+, ClassEval, LeetCode | 923 |
| **Readability** | | |
| Code Length | | 492 |
| Name Convention | HumanEval+ | 984 |
| Comments | | 328 |
| **Maintainability** | | |
| Maintainability Index | ClassEval | 100 |
| Modularity | LeetCode | 540 |
| **Efficiency** | | |
| Time Complexity | LeetCode | 101 |
| Space Complexity | | |

### 3.2 READABILITY

In real-world development scenarios, maintaining a consistent coding style is essential for enhancing comprehensibility and minimizing the time required for code maintenance, often referred to as code readability (Börstler et al., 2023). One of the most fundamental aspects of coding style is line length; excessively long lines can lead to truncation on screens. Additionally, adopting clear and consistent naming conventions enables developers to quickly grasp the functionality of interfaces, while well-

placed comments facilitate a rapid understanding of the implementation logic. Consequently, we condense code readability into three key factors: Length, Naming Convention, and Comment. In response to real-world development needs, we collect a set of customizable options for each factor.

For the **Length** factor, readability requirements for code length can vary depending on display scales in different user scenarios. To address this, we refer to PEP8 style guidelines for Python and define the following user requirements regarding code length: (60, 20), (70, 30), and (79, 40), where the parentheses represent the maximum line length and the maximum number of lines in functions, respectively. For the **Naming Convention** factor, camel-case and snake-case are widely used naming conventions in programming, with specific preferences varying by project. Therefore, we provide the option to choose between camel-case and snake-case based on the conventions employed for functions and variables. For the **Comment** factor, different levels of granularity serve distinct purposes. Line-level comments clarify implementation details and are beneficial for novice programmers, while function-level comments enhance understanding of functionality and usage. Thus, we offer customization options for both comment types. Please kindly note that although a few readability requirements can be addressed using formatting tools, we believe that assessing whether a model can directly generate code that meets readability standards provides valuable insights into the model's ability to follow user instructions. Moreover, we further find that the model's capability in readability serves as a significant indicator of its overall coding proficiency. Refer to Appendix A for corresponding experiment results due to page limitations.

To align with real-world scenarios that require readability, we conduct experiments on HumanEval+ (Liu et al., 2024) dataset, which consists of coding exercise tasks. We incorporate the aforementioned customized requirements into the problem descriptions to evaluate the model's coding capabilities in terms of readability. To measure code readability, we analyze the various components of the generated code using abstract syntax trees. We then develop corresponding rule-based and regular expression-based methods to measure code length, detect naming conventions, and differentiate between different levels of comment granularity.

## 3.3 MAINTAINABILITY

The maintainability of code plays a vital role in the long-term health of software and the efficiency of development teams. Numerous quality models propose empirical quantitative measures to assess maintainability. Additionally, the single responsibility principle is essential in code design, helping to prevent excessive functional coupling. Based on these principles, we identify two key factors influencing code maintainability: Maintainability Metric and Modularity.

For the **Maintainability Metric** factor, we use the Maintainability Index (MI) (Coleman et al., 1994) to measure how maintainable the code is, which is widely used in the Microsoft Visual Studio 2010 development environment. To address concerns regarding the unreliability of evaluations related to the MI metric when faced with differences in the volume and organizational structure of the code under assessment (Heitlager et al., 2007), we conduct assessments solely on fixed evaluation data. We calculate the maintainability index values of the generated code, thereby enabling a horizontal comparison of the complexity of code generated by different models when faced with the same task. This comparison reflects the variations in their ability to produce maintainable code and substantially alleviates the aforementioned concerns.

Specifically, MI is a four-metric polynomial equation, resulting in a value between 0 and 100, with higher values indicating greater maintainability. The formulation is as follows:

$$\text{MI} = \max\left[0, 100 \cdot \frac{171 - 5.2\ln V - 0.23G - 16.2\ln L + 50\sin(\sqrt{2.4C})}{171}\right] \quad (1)$$

where $V$ is Halstead Volume to identify measurable properties of the code, $G$ is Cyclomatic Complexity corresponding to the number of decisions a block of code contains plus 1, $L$ is the number of source lines of code, and $C$ is the percent of comment lines. To comprehensively assess the maintainability requirements satisfaction of LLM-generated code, we conduct experiments on ClassEval (Du et al., 2023) dataset, to ensure the complexity of the code problems.

For the **Modularity** factor, different requirements determine the varying levels of code modularization. Achieving compactness often requires implementing functionality within a single function, while maximizing code reusability typically involves the use of multiple functions. Accordingly,

we define several customization options: implementing functionality using 1, 2, or 3 functions. To assess the modularity of the generated code, we conduct experiments on LeetCode (Guo et al., 2024) dataset, which presents a greater challenge and thus enhances discriminative capability. Additionally, we employ the abstract syntax tree to extract all function nodes from the generated code to verify whether the number of function nodes aligns with the defined level of modularity.

## 3.4 EFFICIENCY

In most applications, code efficiency is closely tied to user experience and business process effectiveness. Typically, efficiency is assessed through time complexity and space complexity. Given the varying hardware conditions of users, it is common practice to balance execution time and memory usage or to optimize one of these aspects to the extreme to ensure code efficiency. To address these scenarios, we gather 101 cases from LeetCode programming problems designed to simulate such conditions. These cases are customized with specific requirements for time complexity, space complexity, or both, to evaluate how well the LLM-generated code meets the efficiency standards.

To measure code efficiency, we propose the Normalized Index (NI), i.e., to measure the degree to which the generated code satisfies the complexity requirement. Given two standard solution with time and space complexity $C_1^T, C_1^S$ and $C_2^T, C_2^S$, respectively, where $C_1^T$ and $C_2^S$ are better, and given their total running time $T_1, T_2$ ($T_1 < T_2$) and memory usage $S_1, S_2$ ($S_1 > S_2$) on all test cases. Now there is a code $\hat{C}$ to be evaluated, which has a running time $\hat{T}$ and memory usage $\hat{S}$, with requirements $C_1^T, C_1^S$, then the normalized index is:

$$\mathrm{NI}_T = 100 \cdot \mathrm{Clip}\left(1 - \frac{\hat{T} - T_1}{T_2 - T_1}, 0, 1\right), \ \mathrm{NI}_S = 100 \cdot \mathrm{Clip}\left(1 - \frac{\hat{S} - S_1}{S_1 - S_2}, 0, 1\right) \quad (2)$$

$\mathrm{NI}_T$ indicates the degree of time complexity toward $C_1^T$, and $\mathrm{NI}_S$ indicates the degree of space complexity toward $C_2^S$.

## 4 EXPERIMENTS

In this section, we conduct a detailed evaluation of 28 LLMs and obtain several valuable findings. We first introduce the input formats and inference configurations for code generation tasks, along with the selection of LLMs. Subsequently, we present the overall experimental findings and conduct further analysis of the results to derive meaningful conclusions. The detailed experimental results are shown in Appendix C.

## 4.1 SETTINGS

**Task formats** We construct the different prompts based on the completion style and chat style, to better induce the LLMs to accomplish the corresponding tasks, see details in Appendix B.2. In the inference process, we use a greedy strategy and set the temperature to 0.

**Models** We select 28 widely-used closed-source and open-source LLMs ranging in different sizes, including state-of-the-art code LLMs. For closed-source models, our experiments include the GPT series (GPT-3.5-turbo-0125, GPT-4o-2024-05-13, and GPT-4o-mini), o1-mini-2024-09-12 (OpenAI, 2024b), and Claude-3.5-Sonnet. For open-source models, our experiments include several series: DeepSeek (Guo et al., 2024; Zhu et al., 2024) (DeepSeek-Coder-Ins-7B/13B/34B and DeepSeek-V2.5), CodeLlama (Roziere et al., 2023) (CodeLlama-Ins-7B/13B/34B and CodeLlama-Python-7B/13B/34B), WizardCoder (Luo et al., 2024) (WizardCoder-15B/33B and WizardCoder-Python-7B/13B), Qwen (Bai et al., 2023; Yang et al., 2024; Hui et al., 2024) (CodeQwen1.5-7B-Chat, Qwen2.5-Coder-7B-Ins, Qwen2-72B-Ins, and Qwen2.5-72B-Ins), Llama3 (Dubey et al., 2024) (Llama3-Ins-8B/70B), Mixtral-8x22B (Jiang et al., 2024) and StarCoder2-15B (Lozhkov et al., 2024). The full list of models is shown in Appendix B.1.

## 4.2 OVERALL RESULTS

The overall evaluation results on all 4 dimensions of each LLM are demonstrated in Table 2, and Figure 3 provides a more intuitive comparison of the capabilities across various dimensions for rep-

Table 2: Based on the RACE benchmark, the performance results for each LLM in code correctness (C), readability (R), maintainability (M), and efficiency (E). RN, RL, RC, and EC denote the Name Convention, Length, Comments, and Complexity factor. MI denotes the Maintainability Index. MC denotes the Modularity factor. $NI_T$ and $NI_S$ are metrics for code efficiency. RACE Score represents the overall metrics at the dimension level. The symbol (*) indicates that the metric is a scalar from 0 to 100, and the rest are percentages (%). The symbol (†) indicates that the results are obtained from a randomly sampled 30% of the evaluation data, in order to optimize budget efficiency.

| Models | RACE Overall | C C | R RN | R RL | R RC | M MI* | M MC | E $NI_T$* | E $NI_S$* |
|---|---|---|---|---|---|---|---|---|---|
| *Instruct-Type* | | | | | | | | | |
| o1-mini-2024-09-12 | **63.5** | **70.1** | **80.7** | 47.5 | **77.7** | 64.4 | **66.1**† | **60.3**† | 40.0† |
| Claude-3.5-Sonnet | 62.3 | 64.6 | 74.4 | 52.0 | 65.5 | 75.3 | 59.8 | 56.8 | **49.7** |
| GPT-4o | 57.2 | 59.9 | 78.6 | 63.2 | 70.4 | 75.1 | 35.2 | 44.0 | 42.0 |
| GPT-4o-mini | 52.5 | 56.4 | 67.6 | 55.7 | 72.9 | 73.5 | 23.3 | 40.3 | 39.5 |
| GPT-3.5-turbo-0125 | 43.6 | 44.7 | 51.4 | 46.1 | 47.5 | 80.2 | 18.5 | 27.5 | 36.5 |
| CL-7B-Ins | 23.2 | 23.9 | 17.8 | 23.4 | 22.2 | 71.8 | 7.2 | 8.2 | 8.8 |
| CL-13B-Ins | 26.9 | 24.4 | 22.9 | 23.6 | 29.0 | 82.1 | 7.6 | 10.4 | 16.1 |
| CL-34B-Ins | 24.4 | 26.0 | 21.9 | 17.5 | 10.7 | 73.2 | 8.5 | 14.4 | 13.8 |
| DS-Coder-6.7B-Ins | 39.8 | 39.2 | 45.8 | 46.6 | 50.0 | 79.3 | 8.2 | 27.1 | 30.0 |
| DS-Coder-7B-Ins | 38.9 | 39.9 | 36.8 | 46.0 | 53.7 | 79.6 | 8.9 | 25.1 | 26.8 |
| DS-Coder-33B-Ins | 44.8 | 44.7 | 59.0 | 53.5 | 54.0 | 75.7 | 11.3 | 35.3 | 36.1 |
| DS-Coder-V2-16B-Ins | 48.2 | 50.9 | 41.8 | 57.7 | 47.5 | 78.2 | 19.8 | 40.2 | 47.7 |
| DS-V2.5-236B | 57.1 | 59.0 | 72.2 | 66.1 | 65.8 | 72.9 | 33.9 | 46.4 | 49.5 |
| CodeQwen1.5-7B-Chat | 45.2 | 46.3 | 48.8 | 47.0 | 62.2 | 82.3 | 13.0 | 30.7 | 37.7 |
| Qwen2.5-Coder-7B-Ins | 49.0 | 57.1 | 53.0 | 51.8 | 61.3 | 78.6 | 17.6 | 37.0 | 33.7 |
| Qwen2-72B-Ins | 50.1 | 53.1 | 73.6 | 47.6 | 60.1 | 79.4 | 22.8 | 32.3 | 39.4 |
| Qwen2.5-72B-Ins | 61.3 | 64.1 | 77.2 | 72.1 | 72.8 | 76.7 | 40.4 | 47.9 | 49.4 |
| Mixtral-8x22B | 42.2 | 42.0 | 56.2 | 47.8 | 56.1 | 79.6 | 9.1 | 24.7 | 33.2 |
| Llama3-8B-Ins | 35.2 | 35.6 | 44.3 | 23.6 | 40.0 | 79.8 | 8.1 | 23.5 | 26.9 |
| Llama3-70B-Ins | 47.2 | 44.4 | 66.0 | 47.8 | 54.2 | 79.8 | 25.2 | 29.2 | 42.8 |
| *Completion-Type* | | | | | | | | | |
| CL-7B-Py | 24.0 | 20.4 | 20.9 | 25.8 | 12.5 | 79.4 | 3.7 | 14.3 | 14.4 |
| CL-13B-Py | 25.6 | 21.7 | 23.1 | 30.9 | 24.4 | 78.6 | 2.4 | 13.8 | 14.7 |
| CL-34B-Py | 23.6 | 19.2 | 18.8 | 26.7 | 8.6 | **85.3** | 2.2 | 12.0 | 14.4 |
| WC-Py-7B | 26.2 | 25.2 | 22.8 | 28.0 | 10.1 | 79.3 | 7.2 | 15.3 | 16.7 |
| WC-Py-13B | 29.3 | 26.3 | 23.9 | 33.1 | 30.5 | 78.8 | 8.5 | 16.2 | 19.8 |
| WC-15B | 30.4 | 28.0 | 24.0 | 27.8 | 28.1 | 80.0 | 7.8 | 21.8 | 24.2 |
| WC-33B | 40.8 | 44.4 | 40.9 | 47.6 | 44.8 | 71.2 | 9.3 | 33.9 | 34.9 |
| StarCoder2-15B | 29.2 | 28.5 | 25.8 | 27.9 | 22.0 | 74.2 | 6.1 | 20.6 | 25.1 |

Figure 3: Performance radar charts of several representative LLMs on the RACE benchmark, visually illustrating the capability distribution across 8 detailed factors.

resentative models. We find that **compared to previous correctness-centric benchmarks, RACE can provide a multi-dimensional comprehensive evaluation for code LLMs, offering valuable insights for their application in real-world scenarios.**

- From an overview perspective, current code LLMs still have considerable room for improvement in generating correct and user-compliant code across multiple dimensions. For instance, even the most advanced model, o1-mini, achieves only a score of 60.3 in time complexity, with most models below 50. Additionally, apart from o1-mini and Claude-3.5-Sonnet, all other models exhibit performance below 45% in modularity. Further, we find that incorporating different user demands into instructions has varying impacts on code accuracy. For example, increasing the requirement to add comments in appropriate sections can enhance accuracy, which we hypothesize is due to comments facilitating an implicit chain-of-thought. In comparison, adding requirements related to code length tends to decrease accuracy, which may be attributed to the model's inherent preferences towards code of varying lengths (see detailed results in Appendix C.1). These findings offer valuable insights for designing better prompting methods and future optimization directions.

Table 3: The performance comparison of LLMs trained on clean data versus contamination data over the same number of epochs.

| Benchmark | 2 Epochs | | | 4 Epochs | | | 6 Epochs | | | 8 Epochs | | | 10 Epochs | | |
|---|---|---|---|---|---|---|---|---|---|---|---|---|---|---|---|
| | Clean | w/ Test | Δ ↓ | Clean | w/ Test | Δ ↓ | Clean | w/ Test | Δ ↓ | Clean | w/ Test | Δ ↓ | Clean | w/ Test | Δ ↓ |
| HumanEval+ | 22.0 | 40.9 | +18.9 | 23.8 | 62.2 | +38.4 | 20.1 | 89.6 | +69.5 | 22.0 | 95.7 | +73.7 | 22.6 | 97.6 | +75.0 |
| MBPP+ | 35.7 | 52.4 | +16.7 | 36.8 | 77.2 | +40.4 | 33.1 | 91.5 | +58.4 | 32.3 | 98.1 | +65.8 | 33.1 | 98.4 | +65.3 |
| ClassEval | 14.0 | 22.0 | +8.0 | 11.0 | 46.0 | +35.0 | 13.0 | 74.0 | +61.0 | 14.0 | 86.0 | +72.0 | 13.0 | 89.0 | +76.0 |
| LeetCode | 3.9 | 3.9 | +0.0 | 7.8 | 20.0 | +12.2 | 6.1 | 70.0 | +63.9 | 7.2 | 95.6 | +88.4 | 7.2 | 97.8 | +90.6 |
| RACE - *Overall* | 20.3 | 15.6 | -4.7 | 18.8 | 20.7 | +1.9 | 19.3 | 35.8 | +16.5 | 19.8 | 42.9 | +23.1 | 19.1 | 43.8 | +24.7 |
| RACE - *IF rate* | 51.1 | 38.3 | **-12.8** | 53.9 | 37.4 | **-16.5** | 53.0 | 40.7 | **-12.3** | 53.4 | 41.9 | **-11.5** | 53.2 | 43.2 | **-10.0** |

- Since current benchmarks use correctness as the sole guiding indicator, some LLMs perform well only on correctness but exhibit significant deficiencies in other dimensions. For example, Qwen2.5-Coder-7B-Ins demonstrates comparable levels of code correctness to GPT-4o-mini; however, GPT-4o-mini outperforms it by at least 5 percentage points regarding comments, modularity, and space complexity. These findings highlight the shortcomings of previous benchmarks and suggest that such deficiencies may be related to potential data leakage (see analysis in Section 4.3).

- The evaluation results reveal the importance of preserving the general instruction-following and language-understanding capabilities of code LLMs during training. This aligns with the direction of recent advancements and provides valuable insights for guiding further development. Specifically, on the one hand, current LLMs with the best coding abilities are often general-purpose LLMs, such as Claude-3.5-Sonnet, GPT-4o, and Qwen2.5-72B-Ins, all with overall scores exceeding 57. On the other hand, in the technical reports for Qwen2.5 (Qwen, 2024) and DeepSeek-Coder-V2 (Zhu et al., 2024), it is mentioned that a significant proportion of natural language corpora and general instruction data are included in training data. This approach not only enhances coding capabilities but also preserves general-purpose abilities. Moreover, both models achieve an overall score exceeding 57, outperforming most code LLMs.

These findings indicate that future research should prioritize improving instruction-following capabilities in terms of code readability, maintainability, and efficiency, while ensuring code accuracy remains uncompromised. This approach seeks to develop code LLMs that consistently meet real-world development requirements across multiple dimensions.

### 4.3 ROBUSTNESS OF RACE ON DATA CONTAMINATION

Data contamination refers to the mixing of evaluation data into the training dataset, resulting in an overly optimistic estimation of the model's true performance and leading to erroneous conclusions. With the rapid expansion in training datasets for LLMs and the opacity surrounding critical information such as data sources and detailed data processing methods, addressing data contamination is crucial for obtaining accurate and trustworthy evaluation results for LLMs. In this regard, RACE plays a vital role, as it requires LLMs to generate code that is correct and meets user-specific customization demands across several dimensions. A contaminated model would only fit the data itself, merely enhancing accuracy without improving the model's ability to follow user instructions. Therefore, it is intuitive to suggest that RACE can effectively mitigate the impact of data contamination.

To validate the RACE benchmark's robustness against data contamination, we select `starcoderbase-7b` (Li et al., 2023a) as our baseline. This model has been carefully curated to exclude data from HumanEval (Chen et al., 2021) and MBPP (Austin et al., 2021) during its training, and ClassEval (Du et al., 2023) and LeetCode (Guo et al., 2024) are not within the temporal coverage of its training data. Furthermore, we compare the performance of models under clean data conditions and varying levels of data contamination on the existing benchmarks and the RACE benchmark. Specifically, we employ LoRA (Hu et al., 2022) to train models on both clean and con-

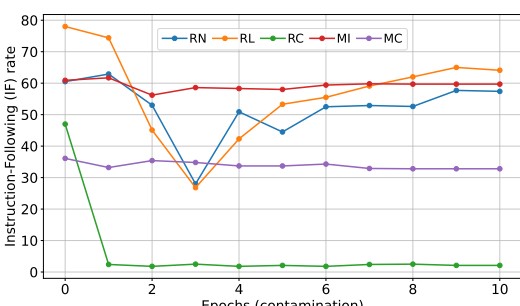

Figure 4: The variation in instruction-following capabilities under different factors in the context of data contamination.

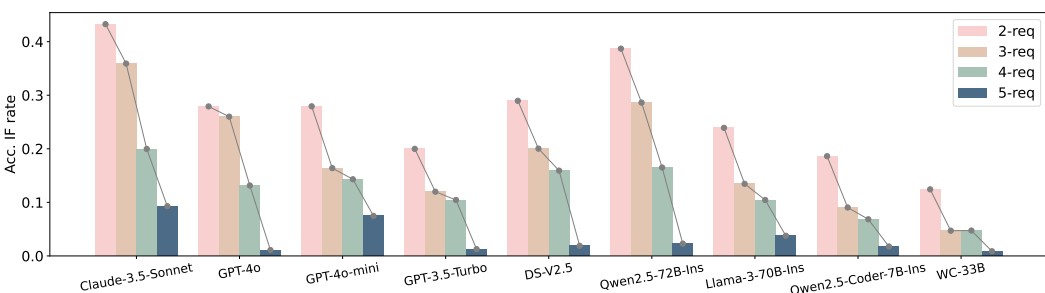

Figure 5: The variation in the ability of different LLMs to generate code that is correct and satisfies all requirements as the number of requirements in complex instructions gradually increases from 2 (2-req) to 5 (5-req).

taminated datasets. In this case, the contaminated dataset consists of HumanEval+, MBPP+, ClassEval, and LeetCode data, while the clean dataset consists of an equivalent number of samples randomly selected from Magicoder-OSS-Instruct (Wei et al., 2023). The model is trained for 10 epochs on the corresponding dataset, with a batch size of 32 and a learning rate of 1e-3. Additionally, `starcoderbase-7b` is pre-trained for 3 epochs on Code-Alpaca (Chaudhary, 2023) to enhance instruction-following before performance comparisons.

The results presented in Table 3 clearly illustrate the role of the RACE benchmark in mitigating the impact of data contamination on evaluation: 1) **The original benchmarks are significantly affected by severe data contamination.** As the level of contamination increases, the code accuracy on the corresponding benchmark rapidly rises. For instance, when trained on the contaminated dataset for 8 epochs, the accuracy of each original benchmark exceeds 85%. 2) **The RACE benchmark provides more stable evaluation results, and therefore effectively resist the risks of data contamination.** When data contamination is present, the instruction-following rate (*IF rate*) of the model on the RACE benchmark consistently remains below 10% compared to the model without contamination. As the degree of contamination increases, the rate of increase in the IF rate significantly slows down. This phenomenon occurs because data contamination merely guides models to fit the data itself, thereby improving the accuracy of generated code, which leads to a slow increase in the proportion of generated code that is both correct and meets user requirements (*Overall score*). However, this does not contribute to improving the model's ability to follow user instructions. Therefore, the RACE benchmark featured multidimensional evaluation effectively mitigates the risks associated with data contamination. Furthermore, Figure 4 illustrates the variation in the model's instruction-following capability across different factors as the degree of data contamination increases. It is evident that **data contamination significantly impairs the model's instruction-following ability in all factors**. For instance, in the case of code comments (RC), the model experiences a dramatic drop from 47% to 2.4% after only one epoch of data leakage. Notably, for factors related to naming conventions (RN) and code length (RC), the model's performance first declines significantly and then gradually improves, ultimately remaining considerably lower than its actual instruction-following capability.

## 4.4 COMPLEX INSTRUCTION FOLLOWING ABILITIES OF CODE LLMs

In real-world development scenarios, the requirements involved are often multifaceted. For example, when developing a real-time data processing system, it is necessary to maintain code efficiency while also ensuring the readability of the code through standardized variable names and comments. To investigate the performance of code LLMs under complex requirements, we construct complex instructions based on LeetCode cases. Specifically, we randomly select $K$ (where $2 \leq K \leq 5$) factors from the five factors: naming convention, length, comment, modularity, and efficiency, to construct complex instructions with $K$ customized requirements. Subsequently, we calculate the proportion of generated code that is correct and meets all customized requirements (Acc.IF), serving as the performance metric for the model's adherence to complex instructions.

The results from representative LLMs are illustrated in Figure 5. It is evident that as the number of requirements increases, the performance of all models gradually declines. **Even the most advanced LLMs struggle to adequately satisfy all customized requirements within the complex instructions.** For instance, when the number of requirements reaches 5 (5-req), almost all models exhibit a significant drop in Acc.IF rate, approaching 0, revealing a notable performance gap. We

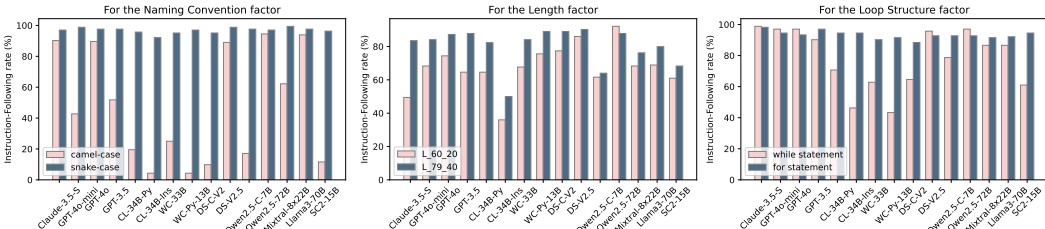

Figure 6: The instruction-following rates of different LLMs for different customization needs. For naming convention, we adapt camel-case or snake-case for both function names and variable names. For code length, we set single-line lengths are limited to 60 and 79 characters, and methods are limited to 20 and 40 lines, respectively. For loop structure, we adapt `for` or `while` statements.

attribute this deficiency to the fact that existing code benchmarks prioritize correctness, overlooking the models' capability to follow complex instructions. It is imperative for future work to investigate the underlying mechanisms of code LLMs when confronted with complex instructions, with the aim of enhancing their applicability in practical development contexts with multifaceted demands.

## 4.5 PREFERENCE BIAS OF CODE LLMs

To investigate whether the internal code preferences of the model affect its ability to follow user instructions, we conduct a more fine-grained comparison across various factors that are likely to induce such preferences. Specifically, we designed distinct instructions for naming convention, code length, and loop structure, respectively. Our objective is to observe whether the model exhibits a stronger capability to meet certain customization requirements. Furthermore, we calculate the proportion of LLM-generated code that follows these customized requirements, referred to as the Instruction-Following (IF) rate.

Figure 6 demonstrates the IF rates of 15 representative LLMs across all the customized requirements above. Our analysis reveals that **most LLMs exhibit an inherent preference bias towards generating code in specific styles.** This bias often leads to difficulties in following user instructions when the requested style diverges from that prevalent in their training data. Specifically, for naming conventions, Python typically employs snake-case for function and variable names. When instructed to use camel-case, most LLMs, such as CodeLlama and WizardCoder, almost fail to comprehend and fulfill this requirement, with IF rates below 30%. Regarding code length, when presented with stricter length constraints, the IF rates of most instruct-type LLMs drop by nearly 15%. When it comes to loop structures, certain LLMs, including CodeLlama-34B-Ins and WizardCoder-33B, exhibit a strong inclination towards using `for` statements. These observations suggest that many LLMs primarily learn the inherent patterns of token prediction from examples, lacking a comprehensive understanding of code logic. Such preference bias may result in the rigidification of coding styles in code LLMs, ultimately impeding their ability to meet specific real-world requirements and affecting the adaptability and scalability of the generated code. This issue could be even more pronounced in programming languages like Perl, JavaScript, and PHP, where there is no strict, widely accepted standard for coding styles.

## 5 CONCLUSION

We introduce the RACE benchmark, a comprehensive multi-dimensional evaluation framework for code generation, including correctness, readability, maintainability, and efficiency. The RACE benchmark assesses the ability of LLMs to generate code that is both correct and meets customized requirements across different dimensions. Through extensive experiments involving 28 representative LLMs, we find that the current code LLMs still fall short in generating high-quality code on demand. Moreover, the RACE benchmark serves as an effective tool for mitigating data contamination. Our research underscores the importance of enhancing code LLMs to generate high-quality code across multiple dimensions beyond mere correctness. Future work should prioritize developing novel learning algorithms to improve the coverage and usability of code LLMs in addressing diverse user needs. This will enable the models to better handle complex instructions, ultimately guiding code LLMs towards becoming practical software development agents.

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

## A  CORRELATION ANALYSIS ACROSS DIMENSIONS

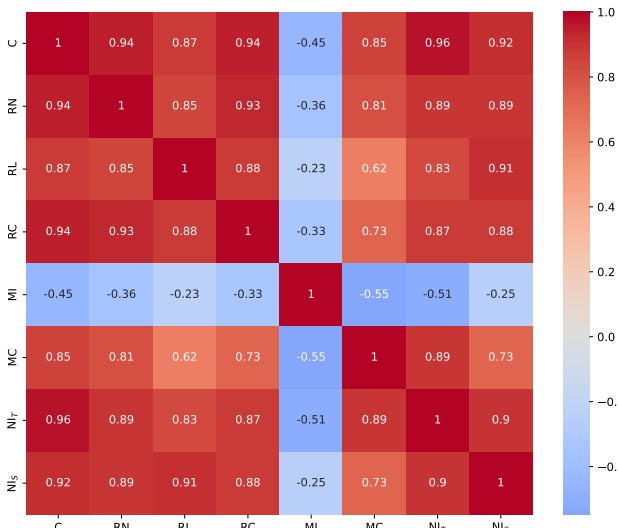

Figure 7: The Pearson correlation coefficient matrix among factors under the dimensions of code correctness, readability, maintainability, and efficiency. We can observe that readability is a critical indicator of overall code quality.

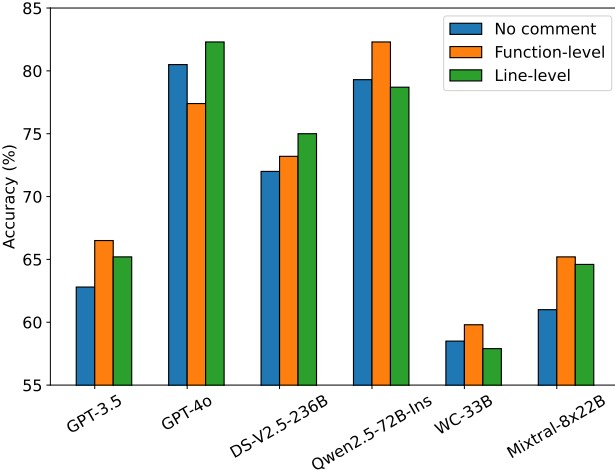

Figure 8: Comparison of code correctness among LLM-generated code without custom requirements, with function-level comments, and with line-level comments.

To conduct a more in-depth analysis of how different factors across various dimensions influence overall code quality, we analyze the correlations between different factors across all models. Specifically, we first compute the proportion of the generated code that is both correct and follows customized instructions across 8 factors for 28 Code LLMs. Subsequently, we calculate the Pearson correlation coefficients between these factors.

The results of the correlation analysis are presented in Figure 7, which demonstrate that **readability serves as a critical indicator of overall code quality**. Notably, significant correlations are observed between readability and nearly all the factors, with most correlation coefficients exceeding 0.8, and particularly surpassing 0.85 in relation to correctness. For example, if LLM-generated code consistently uses proper naming conventions, follows length constraints, and includes appropriate comments, it is more likely to exhibit higher overall quality. This finding aligns with the conclu-

sions of Börstler et al. (2023), which identifies readability as a decisive factor in code quality, suggesting that improving the readability of LLM-generated code is a critical path for enhancement. Furthermore, we analyze the comments factor within the readability dimension and compare the accuracy of LLM-generated code before and after incorporating comments. As shown in Figure 8, requiring models to include comments in appropriate sections of code enhances the performance of some LLMs. We hypothesize that this improvement may be attributed to an emerging ability in large-scale LLMs (Wei et al., 2022; Schaeffer et al., 2024), where comments serve as an implicit chain-of-thought mechanism, thereby enhancing the accuracy of the generated code.

## B EXPERIMENT SETUP

### B.1 MODEL SHORT NAMES

We demonstrate the details of LLMs in our experiment in Table 4.

Table 4: The short names of all LLMs in the experiment.

| Model ID | Short Name |
|---|---|
| claude-3.5-sonnet | Claude-3.5-Sonnet |
| gpt-4o-2024-05-13 | GPT-4o |
| gpt-4o-mini | GPT-4o-mini |
| gpt-3.5-turbo-0125 | GPT-3.5-turbo-0125 |
| o1-mini-2024-09-12 | o1-mini-2024-09-12 |
| CodeLlama-7b-Python-hf | CL-7B-Py |
| CodeLlama-7b-Instruct-hf | CL-7B-Ins |
| CodeLlama-13b-Python-hf | CL-13B-Py |
| CodeLlama-13b-Instruct-hf | CL-13B-Ins |
| CodeLlama-34b-Python-hf | CL-34B-Py |
| CodeLlama-34b-Instruct-hf | CL-34B-Ins |
| WizardCoder-15B-V1.0 | WC-15B |
| WizardCoder-33B-V1.1 | WC-33B |
| WizardCoder-Python-7B-V1.0 | WC-Py-7B |
| WizardCoder-Python-13B-V1.0 | WC-Py-13B |
| deepseek-coder-6.7b-instruct | DS-Coder-6.7B-Ins |
| deepseek-coder-7b-instruct-v1.5 | DS-Coder-7B-Ins |
| deepseek-coder-33b-instruct | DS-Coder-33B-Ins |
| DeepSeek-Coder-V2-Lite-Instruct | DS-Coder-V2-16B-Ins |
| deepseek-v2.5 | DS-V2.5-236B |
| CodeQwen1.5-7B-Chat | CodeQwen1.5-7B-Chat |
| Qwen2.5-Coder-7B-Instruct | Qwen2.5-Coder-7B-Ins |
| Qwen2-72B-Instruct | Qwen2-72B-Ins |
| Qwen2.5-72B-Instruct | Qwen2.5-72B-Ins |
| mixtral-8x22b | Mixtral-8x22B |
| Meta-Llama-3-8B-Instruct | Llama3-8B-Ins |
| Meta-Llama-3-70B-Instruct | Llama3-70B-Ins |
| starcoder2-15b | StarCoder2-15B |

### B.2 EVALUATION DATA AND CUSTOMIZED INSTRUCTIONS

Based on widely recognized data, we design customized requirements that are both reasonable and closely aligned with real-world application scenarios. These requirements are incorporated into the task descriptions to generate evaluation data for our RACE benchmark. Detailed customization instructions for each factor are shown in Figure 9 and Figure 10.

For code correctness, we utilize data from HumanEval+ (Liu et al., 2024), MBPP+ (Liu et al., 2024), ClassEval (Du et al., 2023), and LeetCode (Guo et al., 2024). Code readability is evaluated using HumanEval+ (Liu et al., 2024) data, while maintainability is evaluated using ClassEval (Du et al.,

2023) and LeetCode (Guo et al., 2024) data. Code efficiency is measured using a self-constructed dataset based on LeetCode problems. We follow the task settings defined in the original datasets while incorporating our customization requirements. In the case of the HumanEval+ and MBPP+ datasets (Liu et al., 2024), we modify the original prompt format by extracting the core task descriptions to serve as the final prompts. This adjustment helps prevent conflicts between function template information in the original prompts and our requirements for code readability, providing a more accurate assessment of code-related capabilities. Additionally, it mitigates the potential for data leakage, thereby increasing the difficulty and robustness of the RACE benchmark.

---

**A) The templates for the correctness dimension**

Please generate the Python code to solve the following problem.\n\nProblem:\n\n{problem}

**B) The templates for the readability dimension**

**1) For the Naming Convention factor**

Please generate the Python code to solve the following problem, and use camel case for both function names and variable names.\n\nProblem:\n\n{problem}

Please generate the Python code to solve the following problem, and use snake case for both function names and variable names.\n\nProblem:\n\n{problem}

Please generate the Python code to solve the following problem, and use camel case for function names.\n\nProblem:\n\n{problem}

Please generate the Python code to solve the following problem, and use snake case for function names.\n\nProblem:\n\n{problem}

Please generate the Python code to solve the following problem, and use camel case for variable names.\n\nProblem:\n\n{problem}

Please generate the Python code to solve the following problem, and use snake case for variable names.\n\nProblem:\n\n{problem}

**2) For the Length factor**

Please generate the Python code to solve the following problem, where each line is less than 60 characters long and each function is less than 20 lines long.\n\nProblem:\n\n{problem}

Please generate the Python code to solve the following problem, where each line is less than 70 characters long and each function is less than 30 lines long.\n\nProblem:\n\n{problem}

Please generate the Python code to solve the following problem, where each line is less than 79 characters long and each function is less than 40 lines long.\n\nProblem:\n\n{problem}

**3) For the Comment factor**

Please generate the Python code to solve the following problem, and add the necessary docstring for each function.\n\nProblem:\n\n{problem}

Please generate the Python code to solve the following problem, and add comments for each line in each function.\n\nProblem:\n\n{problem}

---

Figure 9: The prompt templates for each factor in the correctness and readability dimension for the RACE benchmark.

**C) The templates for the maintainability dimension**

**1) For the MI factor**

Please complete the class {class_name} in the following code, and ensure that the code has good maintainability. Code maintainability refers to how easy it is to support and change the code.\n\n"'python\n{skeleton}\n"'

**2) For the Modularity factor**

{problem}\n\nPlease complete the code below to solve above problem, and use only the given function.\n\n{starter_code}

{problem}\n\nPlease complete the code below to solve above problem, and use only the given function and one addition sub-function.\n\n{starter_code}

{problem}\n\nPlease complete the code below to solve above problem, and use only the given function and two addition sub-functions.\n\n{starter_code}

**3) For the loop structure (Only for experiments)**

Please generate the Python code to solve the following problem, and just use the for statement to implement the desired loop structures.\n\nProblem:\n\n{problem}

Please generate the Python code to solve the following problem, and just use the while statement to implement the desired loop structures.\n\nProblem:\n\n{problem}

**D) The templates for the efficiency dimension**

{problem}\n\nPlease complete the code below to solve above problem, and make sure that the time complexity of the code is ${complexity}$.\n\n{starter_code}

{problem}\n\nPlease complete the code below to solve above problem, and make sure that the space complexity of the code is ${complexity}$.\n\n{starter_code}

{problem}\n\nPlease complete the code below to solve above problem, and make sure that the time complexity is ${time\_complexity}$ and the space complexity is ${space\_complexity}$.\n\n{starter_code}

Figure 10: The prompt templates for each factor in the maintainability and efficiency dimension for the RACE benchmark.

## C EXPERIMENTAL RESULTS

### C.1 OVERALL RESULTS WITH DETAILED CODE ACCURACY

The experimental results with the code accuracy before and after incorporating customization instructions are presented in Table 5.

### C.2 DETAILED RESULTS ON CODE READABILITY

The detailed experimental results under all customized instructions for the various readability factors are presented in Table 6 and Table 7. For the Naming Convention factor, we design 6 settings that require generated code to follow specified naming conventions for function names (function_camel, function_snake), variable names (var_camel, var_snake), or both (camel, snake). It is evident that most models struggle to consistently follow the camel-case naming convention. Additionally, the variance in model performance is most pronounced in scenarios requiring function names to follow camel-case conventions (function_camel). For the Length factor, we observe that as the constraints become more stringent, ranging from maximum single-line length of 79 and maximum method line

Table 5: Based on the RACE benchmark, the performance results for each LLM in code correctness (C), readability (R), maintainability (M), and efficiency (E). The performance metrics include accuracy (Acc) (%) and the proportion of code that is both functionally correct and follows customized instructions (Acc. IF) (%). RN, RL, RC, and EC denote the Name Convention, Length, Comments, and Complexity factor. MI denotes the Maintainability Index. MC denotes the Modularity factor. $NI_T$ and $NI_S$ are metrics for code efficiency. RACE Score represents the overall Acc. IF values at the dimension level. The (*) symbol indicates that the indicator is a scalar from 0 to 100, and the rest are percentages (%). The symbol (†) indicates that the results are obtained from a randomly sampled 30% of the evaluation data, in order to optimize budget efficiency.

| | RACE | Correctness | Readability | | | | | | | Maintainability | | | | | | Efficiency | | | |
| | - | C | C | RN | | RL | | RC | | C | MI | | C | MC | | C | EC | | |
| Models | Overall | Acc. | Acc. | Acc. | Acc. IF | Acc. | Acc. IF | Acc. | Acc. IF | Acc. | Acc. | MI* | Acc. | Acc. | Acc. IF | Acc. | Acc. | $NI_T$* | $NI_S$* |
|---|---|---|---|---|---|---|---|---|---|---|---|---|---|---|---|---|---|---|---|
| *Instruct-Type* | | | | | | | | | | | | | | | | | | | |
| Claude-3.5-Sonnet | 62.3 | 64.6 | 77.4 | 76.3 | 74.4 | 62.2 | 52.0 | 74.1 | 65.5 | **42.0** | 32.0 | 75.3 | 71.7 | 68.5 | 59.8 | 68.3 | 66.3 | 56.8 | **49.7** |
| GPT-4o | 57.2 | 59.9 | 80.5 | 81.2 | 78.6 | 78.9 | 63.2 | 79.8 | 70.4 | 38.0 | 35.0 | 75.1 | 57.2 | 56.3 | 35.2 | 59.4 | 58.4 | 44.0 | 42.0 |
| GPT-4o-mini | 52.5 | 56.4 | 78.0 | 76.4 | 67.6 | 70.3 | 55.7 | 74.1 | 72.9 | 37.0 | 27.0 | 73.5 | 51.7 | 49.1 | 23.3 | 52.5 | 46.5 | 40.3 | 39.5 |
| GPT-3.5-turbo-0125 | 43.6 | 44.7 | 62.8 | 63.2 | 51.4 | 60.4 | 46.1 | 65.8 | 47.5 | 28.0 | 24.0 | 80.2 | 31.1 | 28.1 | 18.5 | 39.6 | 32.7 | 27.5 | 36.5 |
| o1-mini-2024-09-12 | **63.5** | **70.1** | 82.9 | 83.2 | 80.7 | 76.4 | 47.5 | 80.2 | 77.7 | 36.0 | 25.0 | 64.4 | 79.6† | 83.3† | 66.1† | 87.1† | 77.4† | 60.3† | 40.0† |
| CL-7B-Ins | 23.2 | 23.9 | 32.3 | 31.5 | 17.8 | 31.7 | 23.4 | 30.2 | 22.2 | 16.0 | 15.0 | 71.8 | 12.2 | 10.9 | 7.2 | 15.8 | 13.9 | 8.2 | 8.8 |
| CL-13B-Ins | 26.9 | 24.4 | 36.0 | 37.7 | 22.9 | 35.0 | 23.6 | 35.7 | 29.0 | 17.0 | 19.0 | 82.1 | 10.6 | 13.1 | 7.6 | 17.8 | 17.8 | 10.4 | 16.1 |
| CL-34B-Ins | 24.4 | 26.0 | 36.0 | 36.5 | 21.9 | 35.8 | 17.5 | 36.3 | 10.7 | 12.0 | 18.0 | 73.2 | 15.6 | 14.2 | 8.5 | 20.8 | 15.8 | 14.4 | 13.8 |
| DS-Coder-6.7B-Ins | 39.8 | 39.2 | 65.2 | 65.5 | 45.8 | 61.2 | 46.6 | 61.2 | 50.0 | 26.0 | 25.0 | 79.3 | 18.9 | 18.7 | 8.2 | 28.7 | 30.7 | 27.1 | 30.0 |
| DS-Coder-7B-Ins | 38.9 | 39.9 | 61.0 | 61.5 | 36.8 | 62.6 | 46.0 | 62.8 | 53.7 | 23.0 | 24.0 | 79.6 | 23.3 | 20.9 | 8.9 | 32.7 | 27.7 | 25.1 | 26.8 |
| DS-Coder-33B-Ins | 44.8 | 44.7 | 65.9 | 64.6 | 59.0 | 65.0 | 53.5 | 66.5 | 54.0 | 28.0 | 30.0 | 75.7 | 22.2 | 27.6 | 11.3 | 45.5 | 38.6 | 35.3 | 36.1 |
| DS-Coder-V2-16B-Ins | 48.2 | 50.9 | 72.0 | 71.2 | 41.8 | 66.5 | 57.7 | 67.1 | 47.5 | 26.0 | 30.0 | 78.2 | 44.4 | 44.3 | 19.8 | 49.5 | 55.4 | 40.2 | 47.7 |
| DS-V2.5-236B | 57.1 | 59.0 | 72.0 | 74.5 | 72.2 | 72.8 | 66.1 | 74.1 | 65.8 | 41.0 | **36.0** | 72.9 | 61.7 | 59.1 | 33.9 | 57.4 | 54.5 | 46.4 | 49.5 |
| CodeQwen1.5-7B-Chat | 45.2 | 46.3 | 76.2 | 76.8 | 48.8 | 73.4 | 47.0 | 74.7 | 62.2 | 22.0 | 22.0 | 82.3 | 33.3 | 32.6 | 13.0 | 39.6 | 38.6 | 30.7 | 37.7 |
| Qwen2.5-Coder-7B-Ins | 49.0 | 57.1 | 78.0 | 81.4 | 53.0 | 77.4 | 51.8 | 75.3 | 61.3 | 29.0 | 27.0 | 78.6 | 54.4 | 50.4 | 17.6 | 59.4 | 48.5 | 37.0 | 33.7 |
| Qwen2-72B-Ins | 50.1 | 53.1 | 73.2 | 76.8 | 73.6 | 74.8 | 47.6 | 71.1 | 60.1 | 40.0 | 33.0 | 79.4 | 42.8 | 37.2 | 22.8 | 45.5 | 40.6 | 32.3 | 39.4 |
| Qwen2.5-72B-Ins | 61.3 | 64.1 | 79.3 | 79.6 | 77.2 | 77.4 | 72.1 | 80.5 | 72.8 | 34.0 | 32.0 | 76.7 | 72.8 | 71.8 | 40.4 | 68.3 | 69.3 | 47.9 | 49.4 |
| Mixtral-8x22B | 42.2 | 42.0 | 61.0 | 64.4 | 56.2 | 62.4 | 47.8 | 64.9 | 56.1 | 33.0 | 30.0 | 79.6 | 20.0 | 22.6 | 9.1 | 35.6 | 31.7 | 24.7 | 33.2 |
| Llama3-8B-Ins | 35.2 | 35.6 | 49.4 | 45.5 | 44.3 | 28.7 | 23.6 | 48.1 | 40.0 | 24.0 | 19.0 | 79.8 | 20.6 | 19.1 | 8.1 | 33.7 | 31.7 | 23.5 | 26.9 |
| Llama3-70B-Ins | 47.2 | 44.4 | 65.2 | 67.8 | 66.0 | 56.1 | 47.8 | 64.6 | 54.2 | 28.0 | 29.0 | 79.8 | 31.7 | 31.7 | 25.2 | 38.6 | 38.6 | 29.2 | 42.8 |
| *Completion-Type* | | | | | | | | | | | | | | | | | | | |
| CL-7B-Py | 24.0 | 20.4 | 29.3 | 29.5 | 20.9 | 30.1 | 25.8 | 24.7 | 12.5 | 11.0 | 10.0 | 79.4 | 5.6 | 6.5 | 3.7 | 14.9 | 15.8 | 14.3 | 14.4 |
| CL-13B-Py | 25.6 | 21.7 | 40.2 | 35.0 | 23.1 | 34.8 | 30.9 | 30.2 | 24.4 | 16.0 | 15.0 | 78.6 | 6.1 | 4.8 | 2.4 | 16.8 | 17.8 | 13.8 | 14.7 |
| CL-34B-Py | 23.6 | 19.2 | 31.7 | 27.2 | 18.8 | 32.5 | 26.7 | 27.8 | 8.6 | 3.0 | 2.0 | 85.3 | 7.2 | 5.4 | 2.2 | 17.8 | 11.9 | 12.0 | 14.4 |
| WC-Py-7B | 26.2 | 25.2 | 34.8 | 35.8 | 22.8 | 34.3 | 28.0 | 35.4 | 10.1 | 19.0 | 23.0 | 79.3 | 10.6 | 9.8 | 7.2 | 19.8 | 19.8 | 15.3 | 16.7 |
| WC-Py-13B | 29.3 | 26.3 | 36.0 | 38.2 | 23.9 | 38.4 | 33.1 | 43.6 | 30.5 | 20.0 | 21.0 | 78.8 | 12.8 | 12.8 | 8.5 | 20.8 | 18.8 | 16.2 | 19.8 |
| WC-15B | 30.4 | 28.0 | 38.4 | 38.7 | 24.0 | 41.9 | 27.8 | 40.0 | 28.1 | 22.0 | 21.0 | 80.0 | 11.7 | 11.5 | 7.8 | 21.8 | 22.8 | 21.8 | 24.2 |
| WC-33B | 40.8 | 44.4 | 58.5 | 58.8 | 40.9 | 62.2 | 47.6 | 58.8 | 44.8 | 34.0 | 34.0 | 71.2 | 26.1 | 25.0 | 9.3 | 38.6 | 35.6 | 33.9 | 34.9 |
| StarCoder2-15B | 29.2 | 28.5 | 36.0 | 39.5 | 25.8 | 40.2 | 27.9 | 35.4 | 22.0 | 24.0 | 25.0 | 74.2 | 16.1 | 13.7 | 6.1 | 26.7 | 25.7 | 20.6 | 25.1 |

count of 40 (L_79_40), to maximum single-line length of 60 and maximum method line count of 20 (L_60_20), most models exhibit a significant decline in their ability to meet requirements. For the Comment factor, models show varying responses to comment-related requirements. However, we find that several models, such as DS-Coder-33B-Ins, DS-V2.5-236B, and WC-Py-13B, improve in code correctness when they meet the code comment requirements.

Table 6: Detailed experimental results for the Name Convention factor in the readability dimension on the RACE benchmark.

| | Readability (Naming Convention) | | | | | | | | | | | | | | | | | | | |
|---|---|---|---|---|---|---|---|---|---|---|---|---|---|---|---|---|---|---|---|---|
| | C | camel | | | snake | | | function_camel | | | function_snake | | | var_camel | | | var_snake | | |
| Models | Acc. | Acc. | IF | Acc. IF | Acc. | IF | Acc. IF | Acc. | IF | Acc. IF | Acc. | IF | Acc. IF | Acc. | IF | Acc. IF | Acc. | IF | Acc. IF |
| Claude-3.5-Sonnet | 77.4 | 75.6 | 90.2 | 70.1 | 76.8 | 97.0 | 76.2 | 78.7 | 98.8 | 78.7 | 77.4 | 98.8 | 77.4 | 74.4 | 90.9 | 69.5 | 75.0 | 97.0 | 74.4 |
| GPT-4o | **80.5** | **81.7** | 89.6 | 73.8 | 80.5 | 97.6 | **79.9** | **84.1** | 98.8 | **83.5** | 79.9 | 99.4 | 79.9 | **81.7** | 90.2 | 75.0 | 79.3 | 98.2 | **79.3** |
| GPT-4o-mini | 78.0 | 73.2 | 42.7 | 34.1 | 76.2 | 98.8 | 76.2 | 75.6 | 86.6 | 65.2 | 78.0 | 99.4 | 78.0 | 79.9 | 95.1 | **77.4** | 75.6 | **99.4** | 75.0 |
| GPT-3.5-turbo-0125 | 62.8 | 65.2 | 51.8 | 37.8 | 61.0 | 97.6 | 59.8 | 63.4 | 87.2 | 56.1 | 62.8 | 98.8 | 62.8 | 64.0 | 41.5 | 30.5 | 62.8 | 98.2 | 61.6 |
| CL-7B-Py | 29.3 | 28.0 | 18.3 | 4.9 | 29.9 | 98.8 | 29.9 | 29.9 | 22.0 | 7.3 | 31.7 | 98.2 | 31.7 | 28.7 | 78.7 | 23.2 | 28.7 | 98.2 | 28.7 |
| CL-7B-Ins | 32.3 | 31.7 | 2.4 | 0.0 | 29.3 | 98.2 | 29.3 | 31.1 | 5.5 | 0.0 | 31.7 | **100.0** | 31.7 | 31.7 | 43.9 | 12.2 | 33.5 | **99.4** | 33.5 |
| CL-13B-Py | 40.2 | 34.8 | 6.7 | 2.4 | 35.4 | 97.0 | 35.4 | 36.0 | 9.1 | 3.7 | 33.5 | 97.6 | 33.5 | 34.8 | 73.2 | 28.0 | 35.4 | 98.2 | 35.4 |
| CL-13B-Ins | 36.0 | 37.2 | 4.3 | 3.0 | 37.2 | **99.4** | 37.2 | 40.9 | 9.8 | 5.5 | 34.8 | 99.4 | 34.8 | 40.2 | 48.8 | 20.7 | 36.0 | **99.4** | 36.0 |
| CL-34B-Py | 31.7 | 27.4 | 19.5 | 5.5 | 28.0 | 95.7 | 27.4 | 29.9 | 21.3 | 6.1 | 26.2 | 97.6 | 26.2 | 26.2 | 80.5 | 22.6 | 25.6 | 97.0 | 25.0 |
| CL-34B-Ins | 36.0 | 37.2 | 4.3 | 2.4 | 34.8 | 92.1 | 34.8 | 36.6 | 5.5 | 2.4 | 36.6 | 97.0 | 36.6 | 37.8 | 47.6 | 19.5 | 36.0 | 94.5 | 36.0 |
| WC-15B | 38.4 | 39.6 | 4.3 | 1.2 | 40.9 | 98.2 | 40.9 | 38.4 | 5.5 | 1.2 | 38.4 | 97.6 | 38.4 | 39.0 | 62.8 | 26.2 | 36.0 | 97.6 | 36.0 |
| WC-33B | 58.5 | 57.9 | 25.0 | 14.6 | 59.1 | 95.1 | 57.3 | 57.3 | 34.1 | 20.7 | 57.9 | 97.6 | 57.9 | 59.8 | 60.4 | 35.4 | 61.0 | 95.7 | 59.8 |
| WC-Py-7B | 34.8 | 34.8 | 4.9 | 1.8 | 34.1 | 95.7 | 34.1 | 34.8 | 5.5 | 1.2 | 34.1 | 97.6 | 34.1 | 37.8 | 62.8 | 26.8 | 39.0 | 94.5 | 39.0 |
| WC-Py-13B | 36.0 | 38.4 | 4.3 | 1.8 | 36.6 | 97.0 | 36.6 | 36.6 | 6.1 | 1.2 | 38.4 | 97.6 | 38.4 | 37.8 | 59.8 | 23.8 | 41.5 | 96.3 | 41.5 |
| DS-Coder-6.7B-Ins | 65.2 | 65.2 | 26.2 | 15.9 | 65.9 | 97.6 | 64.6 | 67.7 | 47.0 | 29.9 | 67.7 | **100.0** | 67.7 | 62.8 | 48.2 | 33.5 | 64.0 | 98.2 | 63.4 |
| DS-Coder-7B-Ins | 61.0 | 61.6 | 9.1 | 6.1 | 59.1 | **99.4** | 58.5 | 62.2 | 11.6 | 7.3 | 61.6 | **100.0** | 61.6 | 62.8 | 43.9 | 26.2 | 61.6 | 98.8 | 61.0 |
| DS-Coder-33B-Ins | 65.9 | 64.6 | 73.2 | 51.2 | 65.2 | 97.0 | 64.6 | 62.2 | **99.4** | 61.6 | 64.0 | **100.0** | 64.0 | 68.3 | 73.2 | 50.0 | 63.4 | 97.6 | 62.8 |
| DS-Coder-V2-16B-Ins | 72.0 | 72.0 | 9.8 | 7.9 | 69.5 | 95.1 | 67.7 | 72.6 | 14.0 | 10.4 | 71.3 | 99.4 | 71.3 | 73.2 | 33.5 | 26.2 | 68.9 | 95.1 | 67.1 |
| DS-V2.5-236B | 72.0 | 75.0 | 89.0 | 67.7 | 76.2 | 98.8 | 75.6 | 74.4 | 98.8 | 74.4 | 75.0 | 99.4 | 75.0 | 72.0 | 90.9 | 67.1 | 74.4 | 97.6 | 73.2 |
| CodeQwen1.5-7B-Chat | 76.2 | 75.6 | 12.2 | 9.1 | 76.2 | 97.6 | 75.0 | 76.2 | 15.9 | 11.0 | 79.3 | 99.4 | 78.7 | 76.8 | 57.9 | 43.3 | 76.8 | 96.3 | 75.6 |
| Qwen2.5-Coder-7B-Ins | 78.0 | 81.1 | 17.1 | 14.6 | **81.1** | 97.6 | 78.7 | 82.9 | 36.0 | 31.1 | **81.7** | **100.0** | **81.7** | 80.5 | 40.9 | 32.9 | **81.1** | 97.6 | 78.7 |
| Qwen2-72B-Ins | 73.2 | 75.6 | 90.2 | 68.3 | 78.7 | 98.2 | 78.0 | 75.6 | 93.9 | 69.5 | 79.9 | **100.0** | 79.9 | 74.4 | **95.7** | 70.7 | 76.8 | 97.6 | 75.0 |
| Qwen2.5-72B-Ins | 79.3 | 78.7 | **94.5** | **74.4** | 81.1 | 97.0 | 78.0 | 79.9 | 99.4 | 79.3 | 80.5 | **100.0** | 80.5 | 78.0 | 93.3 | 73.8 | 79.3 | 97.6 | 77.4 |
| Mixtral-8x22B | 61.0 | 65.2 | 62.2 | 43.3 | 64.6 | **99.4** | 64.0 | 65.9 | 96.3 | 62.2 | 64.0 | **100.0** | 64.0 | 65.9 | 40.2 | 26.2 | 64.0 | 98.2 | 63.4 |
| Llama3-8B-Ins | 49.4 | 49.4 | 88.4 | 47.0 | 31.7 | 57.9 | 30.5 | 51.8 | 97.6 | 51.8 | 44.5 | 89.6 | 44.5 | 50.6 | 90.2 | 48.8 | 45.1 | 89.6 | 43.3 |
| Llama3-70B-Ins | 65.2 | 70.1 | 93.9 | 65.9 | 65.9 | 97.6 | 64.6 | 68.3 | **99.4** | 68.3 | 66.5 | **100.0** | 66.5 | 68.9 | 91.5 | 64.0 | 67.1 | 97.6 | 66.5 |
| StarCoder2-15B | 36.0 | 41.5 | 11.6 | 4.3 | 38.4 | 96.3 | 38.4 | 38.4 | 14.0 | 5.5 | 38.4 | 96.3 | 38.4 | 42.1 | 72.0 | 30.5 | 38.4 | 95.7 | 37.8 |

Table 7: Detailed experimental results for the Length and Comment factor in the readability dimension on the RACE benchmark.

| | Readability (Length) | | | | | | | | | | Readability (Comment) | | | | | | Maintainability (Loop Structure) | | | | | |
|---|---|---|---|---|---|---|---|---|---|---|---|---|---|---|---|---|---|---|---|---|---|---|
| | C | L_60_20 | | | L_70_30 | | | L_79_40 | | | by_function | | | by_line | | | for | | | while | | |
| Models | Acc. | Acc. | IF | Acc. IF | Acc. | IF | Acc. IF | Acc. | IF | Acc. IF | Acc. | IF | Acc. IF | Acc. | IF | Acc. IF | Acc. | IF | Acc. IF | Acc. | IF | Acc. IF |
| Claude-3.5-Sonnet | 77.4 | 50.0 | 49.4 | 37.2 | 67.7 | 78.0 | 57.9 | 68.9 | 83.5 | 61.0 | 75.0 | 95.7 | 75.0 | 73.2 | 74.4 | 56.1 | 70.1 | **98.2** | 68.3 | 66.5 | **98.8** | 65.9 |
| GPT-4o | **80.5** | **80.5** | 74.4 | 61.6 | 76.2 | 75.0 | 58.5 | 79.9 | 87.2 | 69.5 | 77.4 | 98.2 | 77.4 | **82.3** | 76.8 | 63.4 | 75.0 | 93.3 | 71.3 | 70.1 | 97.0 | 68.9 |
| GPT-4o-mini | 78.0 | 70.7 | 68.3 | 51.8 | 73.2 | 72.0 | 55.5 | 67.1 | 84.1 | 59.8 | 73.8 | 98.8 | 73.8 | 74.4 | **95.1** | **72.0** | 71.3 | 94.5 | 67.1 | 65.2 | 97.0 | 64.6 |
| GPT-3.5-turbo-0125 | 62.8 | 58.5 | 64.6 | 39.0 | 61.0 | 78.0 | 45.7 | 61.6 | 87.8 | 53.7 | 66.5 | 95.1 | 64.0 | 65.2 | 45.1 | 31.1 | 56.7 | 97.0 | 54.9 | 52.4 | 90.2 | 48.8 |
| CL-7B-Py | 29.3 | 29.9 | 68.3 | 23.8 | 31.1 | 78.0 | 25.6 | 29.3 | 83.5 | 28.0 | 28.7 | 72.6 | 22.0 | 20.7 | 43.3 | 3.0 | 28.7 | 93.9 | 26.2 | 26.8 | 58.5 | 13.4 |
| CL-7B-Ins | 32.3 | 29.9 | 50.6 | 20.1 | 31.7 | 57.9 | 23.2 | 33.5 | 70.7 | 26.8 | 29.9 | **100.0** | 29.9 | 30.5 | 52.4 | 14.6 | 31.7 | 95.7 | 31.1 | 29.9 | 42.7 | 11.0 |
| CL-13B-Py | 40.2 | 34.1 | 78.7 | 27.4 | 34.8 | 83.5 | 31.1 | 35.4 | 88.4 | 34.1 | 34.8 | 92.1 | 34.8 | 25.6 | 62.8 | 14.0 | 33.5 | 91.5 | 31.1 | 34.8 | 60.4 | 18.9 |
| CL-13B-Ins | 36.0 | 34.8 | 53.0 | 20.1 | 35.4 | 62.8 | 25.6 | 34.8 | 64.0 | 25.0 | 36.6 | 92.7 | 34.8 | 34.8 | 57.3 | 23.2 | 31.1 | 95.1 | 30.5 | 34.1 | 45.7 | 14.0 |
| CL-34B-Py | 31.7 | 32.3 | 64.6 | 22.0 | 33.5 | 72.6 | 28.7 | 31.7 | 82.3 | 29.3 | 23.2 | 67.7 | 12.8 | 32.3 | 29.9 | 4.3 | 25.6 | 94.5 | 25.0 | 25.6 | 70.7 | 12.2 |
| CL-34B-Ins | 36.0 | 33.5 | 36.0 | 15.9 | 36.6 | 39.0 | 15.9 | 37.2 | 50.0 | 20.7 | 35.4 | 38.4 | 12.8 | 37.2 | 34.1 | 8.5 | 36.0 | 94.5 | 36.0 | 35.4 | 46.3 | 15.2 |
| WC-15B | 38.4 | 42.7 | 50.0 | 20.7 | 40.2 | 67.1 | 28.0 | 42.7 | 77.4 | 34.8 | 41.5 | 99.4 | 41.5 | 38.4 | 30.5 | 14.6 | 42.7 | 97.6 | 41.5 | 40.2 | 59.1 | 21.3 |
| WC-33B | 58.5 | 62.2 | 67.7 | 42.7 | 62.8 | 76.2 | 48.2 | 61.6 | 84.1 | 51.8 | 59.8 | 98.2 | 58.5 | 57.9 | 49.4 | 31.1 | 59.8 | 90.2 | 54.9 | 59.8 | 62.8 | 36.0 |
| WC-Py-7B | 34.8 | 35.4 | 72.6 | 25.6 | 34.1 | 81.1 | 28.0 | 33.5 | 85.4 | 30.5 | 33.5 | 40.9 | 13.4 | 37.2 | 22.6 | 6.7 | 36.0 | 93.9 | 34.1 | 35.4 | 40.9 | 11.6 |
| WC-Py-13B | 36.0 | 40.2 | 75.6 | 32.9 | 37.8 | 84.8 | 31.7 | 37.2 | 89.0 | 34.8 | 43.3 | 43.3 | 17.7 | 43.9 | 37.2 | 17.7 | 43.3 | 91.5 | 38.4 | 39.0 | 43.3 | 16.5 |
| DS-Coder-6.7B-Ins | 65.2 | 62.2 | 61.0 | 40.9 | 61.0 | 76.8 | 47.6 | 60.4 | 82.9 | 51.2 | 64.0 | **100.0** | 64.0 | 58.5 | 56.7 | 36.0 | 64.0 | 90.9 | 59.1 | 62.8 | 65.9 | 39.6 |
| DS-Coder-7B-Ins | 61.0 | 61.6 | 57.3 | 37.2 | 62.2 | 71.3 | 47.0 | 64.0 | 84.1 | 53.7 | 62.2 | 99.4 | 62.2 | 63.4 | 66.5 | 45.1 | 61.6 | 95.1 | 58.5 | 57.9 | 64.0 | 39.6 |
| DS-Coder-33B-Ins | 65.9 | 62.8 | 73.8 | 47.6 | 65.2 | 84.1 | 53.7 | 67.1 | 90.2 | 59.1 | 68.9 | **100.0** | 64.0 | 64.0 | 61.6 | 39.0 | 66.5 | 91.5 | 60.4 | 68.3 | 70.1 | 48.2 |
| DS-Coder-V2-16B-Ins | 72.0 | 66.5 | 77.4 | 53.7 | 65.9 | 84.8 | 57.9 | 67.1 | 89.0 | 61.6 | 67.7 | 98.2 | 67.7 | 66.5 | 43.9 | 27.4 | 70.7 | 88.4 | 62.2 | 63.4 | 64.6 | 38.4 |
| DS-V2.5-236B | 72.0 | 72.0 | 86.0 | 63.4 | 72.0 | 93.3 | 67.1 | 74.4 | 90.2 | 67.7 | 73.2 | 96.3 | 72.6 | 75.0 | 78.7 | 59.1 | 70.1 | 92.7 | 65.9 | 67.7 | 95.7 | 66.5 |
| CodeQwen1.5-7B-Chat | 76.2 | 71.3 | 47.0 | 36.6 | 75.6 | 61.0 | 48.2 | 73.2 | 74.4 | 56.1 | 76.2 | 98.8 | 75.0 | 73.2 | 62.8 | 49.4 | 72.0 | 93.3 | 68.3 | 65.2 | 70.1 | 44.5 |
| Qwen2.5-Coder-7B-Ins | 78.0 | 77.4 | 61.6 | 50.6 | **78.0** | 70.7 | 54.3 | 76.8 | 64.0 | 50.6 | 79.3 | 99.4 | 79.1 | 71.3 | 61.0 | 43.3 | **78.7** | 92.7 | **73.8** | 71.3 | 78.7 | 55.5 |
| Qwen2-72B-Ins | 73.2 | 72.6 | 62.8 | 46.3 | 76.2 | 64.6 | 48.2 | 75.6 | 65.9 | 48.2 | 73.2 | 98.8 | 72.0 | 68.9 | 67.1 | 48.2 | 73.2 | 90.2 | 67.1 | 70.7 | 81.1 | 61.0 |
| Qwen2.5-72B-Ins | 79.3 | 75.6 | **92.1** | **71.3** | 76.2 | **96.3** | **73.8** | **80.5** | 87.8 | **71.3** | 82.3 | 98.8 | 81.7 | 78.7 | 79.9 | 64.0 | 78.0 | 92.7 | 72.0 | **72.6** | 97.0 | **70.7** |
| Mixtral-8x22B | 61.0 | 61.6 | 68.3 | 44.5 | 63.4 | 75.0 | 50.6 | 62.2 | 76.2 | 48.2 | 65.2 | **100.0** | 65.2 | 64.6 | 69.5 | 47.0 | 56.7 | 91.5 | 51.2 | 57.3 | 86.6 | 51.8 |
| Llama3-8B-Ins | 49.4 | 30.5 | 40.9 | 22.0 | 31.1 | 51.8 | 27.4 | 24.4 | 45.1 | 21.3 | 50.0 | 99.4 | 50.0 | 46.3 | 60.4 | 29.9 | 49.4 | 93.9 | 46.3 | 48.8 | 73.8 | 35.4 |
| Llama3-70B-Ins | 65.2 | 57.3 | 68.9 | 43.3 | 56.7 | 78.7 | 49.4 | 54.3 | 79.9 | 50.6 | 65.2 | **100.0** | 65.2 | 64.0 | 69.5 | 43.3 | 65.2 | 92.1 | 61.6 | 59.8 | 86.6 | 52.4 |
| StarCoder2-15B | 36.0 | 39.6 | 61.0 | 23.2 | 42.1 | 68.9 | 29.3 | 39.0 | 68.3 | 31.1 | 34.8 | 89.6 | 34.1 | 36.0 | 29.3 | 9.8 | 36.0 | 94.5 | 34.1 | 35.4 | 61.0 | 15.9 |

# D CASE STUDIES

The tasks provided by our benchmark present challenges that even the most advanced LLMs find difficult to handle. Here, we show some examples with Claude-3.5-Sonnet.

## D.1 CORRECTNESS

**Good Case**: The model successfully generates functionally correct code.

```
# Input
Please generate the Python code to solve the following problem.

Problem:

Input to this function is a string containing multiple groups of nested parentheses. Your goal
    is to
separate those group into separate strings and return the list of those.
Separate groups are balanced (each open brace is properly closed) and not nested within each
    other
Ignore any spaces in the input string.
```

```
# Output
```

```
1080
1081   def separate_parentheses_groups(s):
1082       # Remove all spaces from the input string
       s = s.replace(" ", "")
1083
       result = []
1084       stack = []
       current_group = ""
1085       balance = 0
1086
1087       for char in s:
           if char == '(':
1088               balance += 1
               stack.append(char)
1089               current_group += char
1090           elif char == ')':
               balance -= 1
1091               stack.pop()
               current_group += char
1092
1093               # If balance is 0, we've completed a group
               if balance == 0:
1094                   result.append(current_group)
1095                   current_group = ""
1096
1097       return result
```

**Bad Case**: The model fails to generate functionally correct code.

```
# Input
Please generate the Python code to solve the following problem.

Problem:

You're given a list of deposit and withdrawal operations on a bank account that starts with
zero balance. Your task is to detect if at any point the balance of account fallls below zero,
    and
at that point function should return True. Otherwise it should return False.
```

```
# Output
def check_negative_balance(operations):
    balance = 0
    for operation in operations:
        if operation.startswith("d"):  # deposit
            amount = int(operation[1:])
            balance += amount
        elif operation.startswith("w"):  # withdrawal
            amount = int(operation[1:])
            balance -= amount
            if balance < 0:
                return True
    return False
```

## D.2  READABILITY

**Good Case**: The model successfully generates code with function names employing camel case.

```
# Input
Please generate the Python code to solve the following problem, and use camel case for
    function names.

Problem:

Check if in given list of numbers, are any two numbers closer to each other than
given threshold.
```

```
# Output
def areNumbersCloserThanThreshold(numbers, threshold):
    """
    Check if any two numbers in the list are closer to each other than the given threshold.

    :param numbers: List of numbers to check
    :param threshold: The minimum distance allowed between any two numbers
    :return: True if any two numbers are closer than the threshold, False otherwise
    """
    # Sort the list to make comparisons easier
```

```
sortedNumbers = sorted(numbers)

# Compare adjacent numbers in the sorted list
for i in range(len(sortedNumbers) - 1):
    if abs(sortedNumbers[i] - sortedNumbers[i+1]) < threshold:
        return True

# If we've gone through all pairs without finding any closer than the threshold
return False
```

**Bad Case**: The model fails to generate code with function names employing camel case.

```
# Input
Please generate the Python code to solve the following problem, and use camel case for
    function names.

Problem:

Find the shortest palindrome that begins with a supplied string.
Algorithm idea is simple:
- Find the longest postfix of supplied string that is a palindrome.
- Append to the end of the string reverse of a string prefix that comes before the palindromic
    suffix.

Given the following code:

```python
def is_palindrome(string: str) -> bool:
    """ Test if given string is a palindrome """
    return string == string[::-1]
```
```

```
# Output
def is_palindrome(string: str) -> bool:
    """ Test if given string is a palindrome """
    return string == string[::-1]

def findShortestPalindrome(s: str) -> str:
    """
    Find the shortest palindrome that begins with the supplied string.
    """
    if not s or is_palindrome(s):
        return s

    for i in range(len(s) - 1, -1, -1):
        if is_palindrome(s[:i+1]):
            return s[i+1:][::-1] + s

    # If no palindromic suffix is found, reverse the entire string and append
    return s[::-1][1:] + s
```

**Good Case**: The model successfully generates code accompanied by line-by-line comments.

```
# Input
Please generate the Python code to solve the following problem, and add comments for each line
    in each function.

Problem:

Check if in given list of numbers, are any two numbers closer to each other than
given threshold.
```

```
# Output
def are_numbers_close(numbers, threshold):
    # Sort the list of numbers in ascending order
    sorted_numbers = sorted(numbers)

    # Iterate through the sorted list, comparing adjacent pairs
    for i in range(len(sorted_numbers) - 1):
        # Calculate the difference between adjacent numbers
        difference = sorted_numbers[i+1] - sorted_numbers[i]

        # Check if the difference is less than the threshold
        if difference < threshold:
            # If a pair is found closer than the threshold, return True
            return True
```

```
      # If no pair is found closer than the threshold, return False
```

**Bad Case**: The model fails to generate code accompanied by line-by-line comments.

```
# Input
Please generate the Python code to solve the following problem, and add comments for each line
    in each function.

Problem:

Input are two strings a and b consisting only of 1s and 0s.
Perform binary XOR on these inputs and return result also as a string.
```

```python
# Output
def binary_xor(a: str, b: str) -> str:
    # Function to perform binary XOR on two input strings

    # Ensure both input strings have the same length by padding with zeros
    max_length = max(len(a), len(b))
    a = a.zfill(max_length)
    b = b.zfill(max_length)

    # Initialize an empty result string
    result = ""

    # Iterate through each position in the input strings
    for i in range(max_length):
        # Perform XOR on corresponding bits and add to result
        if a[i] == b[i]:
            result += "0"
        else:
            result += "1"

    # Remove leading zeros from the result
    result = result.lstrip("0")

    # Return "0" if result is empty (all zeros), otherwise return result
    return result if result else "0"
```

**Good Case**: The model successfully generates code where each line is less than 70 characters long and each function is less than 30 lines long.

```
# Input
Please generate the Python code to solve the following problem, where each line is less than
    70 characters long and each function is less than 30 lines long.

Problem:

Check if in given list of numbers, are any two numbers closer to each other than
given threshold.
```

```python
# Output
def are_numbers_closer_than_threshold(numbers, threshold):
    if len(numbers) < 2:
        return False

    sorted_numbers = sorted(numbers)

    for i in range(len(sorted_numbers) - 1):
        if abs(sorted_numbers[i] - sorted_numbers[i+1]) < threshold:
            return True

    return False
```

**Bad Case**: The model fails to generate code where each line is less than 70 characters long and each function is less than 30 lines long.

```
# Input
Please generate the Python code to solve the following problem, where each line is less than
    70 characters long and each function is less than 30 lines long.

Problem:

This function takes a list l and returns a list l' such that
l' is identical to l in the odd indicies, while its values at the even indicies are equal
to the values of the even indicies of l, but sorted.
```

```
# Output
def sort_even_indices(l):
    # Extract values at even indices
    even_values = [l[i] for i in range(0, len(l), 2)]

    # Sort the even values
    sorted_even = sorted(even_values)

    # Create a new list with sorted even values and original odd values
    result = []
    even_index = 0

    for i in range(len(l)):
        if i % 2 == 0:
            result.append(sorted_even[even_index])
            even_index += 1
        else:
            result.append(l[i])

    return result
```

## D.3 MAINTAINABILITY

**Good Case**: The model successfully generates code with only the given function and two addition sub-functions.

```
# Input
You are given a string word containing distinct lowercase English letters.
Telephone keypads have keys mapped with distinct collections of lowercase English letters,
    which can be used to form words by pushing them. For example, the key 2 is mapped with ["
    a","b","c"], we need to push the key one time to type "a", two times to type "b", and
    three times to type "c" .
It is allowed to remap the keys numbered 2 to 9 to distinct collections of letters. The keys
    can be remapped to any amount of letters, but each letter must be mapped to exactly one
    key. You need to find the minimum number of times the keys will be pushed to type the
    string word.
Return the minimum number of pushes needed to type word after remapping the keys.
An example mapping of letters to keys on a telephone keypad is given below. Note that 1, *, #,
    and 0 do not map to any letters.

Example 1:

Input: word = "abcde"
Output: 5
Explanation: The remapped keypad given in the image provides the minimum cost.
"a" -> one push on key 2
"b" -> one push on key 3
"c" -> one push on key 4
"d" -> one push on key 5
"e" -> one push on key 6
Total cost is 1 + 1 + 1 + 1 + 1 = 5.
It can be shown that no other mapping can provide a lower cost.

Example 2:

Input: word = "xycdefghij"
Output: 12
Explanation: The remapped keypad given in the image provides the minimum cost.
"x" -> one push on key 2
"y" -> two pushes on key 2
"c" -> one push on key 3
"d" -> two pushes on key 3
"e" -> one push on key 4
"f" -> one push on key 5
"g" -> one push on key 6
"h" -> one push on key 7
"i" -> one push on key 8
"j" -> one push on key 9
Total cost is 1 + 2 + 1 + 2 + 1 + 1 + 1 + 1 + 1 + 1 = 12.
It can be shown that no other mapping can provide a lower cost.

Constraints:
```

```
1 <= word.length <= 26
word consists of lowercase English letters.
All letters in word are distinct.

Please complete the code below to solve above problem, and use only the given function and two
    addition sub-functions:
```python
class Solution:
    def minimumPushes(self, word: str) -> int:

```
```

```
# Output
from collections import Counter

class Solution:
    def minimumPushes(self, word: str) -> int:
        def count_frequencies(word):
            return Counter(word)

        def calculate_pushes(frequencies):
            total_pushes = 0
            for i, freq in enumerate(frequencies):
                pushes = (i // 8) + 1
                total_pushes += pushes * freq
            return total_pushes

        freq_count = count_frequencies(word)
        sorted_frequencies = sorted(freq_count.values(), reverse=True)
        return calculate_pushes(sorted_frequencies)
```

**Bad Case**: The model fails to generate code with only the given function and two addition sub-functions.

```
# Input
You are given three positive integers n, x, and y.
In a city, there exist houses numbered 1 to n connected by n streets. There is a street
    connecting the house numbered i with the house numbered i + 1 for all 1 <= i <= n - 1 .
    An additional street connects the house numbered x with the house numbered y.
For each k, such that 1 <= k <= n, you need to find the number of pairs of houses (house1,
    house2) such that the minimum number of streets that need to be traveled to reach house2
    from house1 is k.
Return a 1-indexed array result of length n where result[k] represents the total number of
    pairs of houses such that the minimum streets required to reach one house from the other
    is k.
Note that x and y can be equal.

Example 1:

Input: n = 3, x = 1, y = 3
Output: [6,0,0]
Explanation: Let's look at each pair of houses:
- For the pair (1, 2), we can go from house 1 to house 2 directly.
- For the pair (2, 1), we can go from house 2 to house 1 directly.
- For the pair (1, 3), we can go from house 1 to house 3 directly.
- For the pair (3, 1), we can go from house 3 to house 1 directly.
- For the pair (2, 3), we can go from house 2 to house 3 directly.
- For the pair (3, 2), we can go from house 3 to house 2 directly.

Example 2:

Input: n = 5, x = 2, y = 4
Output: [10,8,2,0,0]
Explanation: For each distance k the pairs are:
- For k == 1, the pairs are (1, 2), (2, 1), (2, 3), (3, 2), (2, 4), (4, 2), (3, 4), (4, 3),
    (4, 5), and (5, 4).
- For k == 2, the pairs are (1, 3), (3, 1), (1, 4), (4, 1), (2, 5), (5, 2), (3, 5), and (5, 3)
    .
- For k == 3, the pairs are (1, 5), and (5, 1).
- For k == 4 and k == 5, there are no pairs.

Example 3:

Input: n = 4, x = 1, y = 1
```

```
Output: [6,4,2,0]
Explanation: For each distance k the pairs are:
- For k == 1, the pairs are (1, 2), (2, 1), (2, 3), (3, 2), (3, 4), and (4, 3).
- For k == 2, the pairs are (1, 3), (3, 1), (2, 4), and (4, 2).
- For k == 3, the pairs are (1, 4), and (4, 1).
- For k == 4, there are no pairs.

Constraints:

2 <= n <= 100
1 <= x, y <= n

Please complete the code below to solve above problem, and use only the given function and two
    addition sub-functions:
```python
class Solution:
    def countOfPairs(self, n: int, x: int, y: int) -> List[int]:

```
```

```python
# Output
class Solution:
    def countOfPairs(self, n: int, x: int, y: int) -> List[int]:
        # Initialize the distance matrix
        dist = self.initializeDistanceMatrix(n, x, y)

        # Apply Floyd-Warshall algorithm
        self.floydWarshall(dist, n)

        # Count pairs for each distance
        return self.countPairsForEachDistance(dist, n)

    def initializeDistanceMatrix(self, n: int, x: int, y: int) -> List[List[int]]:
        # Initialize distance matrix with infinity
        dist = [[float('inf')] * n for _ in range(n)]

        # Set distance to self as 0
        for i in range(n):
            dist[i][i] = 0

        # Set distance for adjacent houses
        for i in range(n - 1):
            dist[i][i+1] = dist[i+1][i] = 1

        # Set distance for the additional street
        if x != y:
            dist[x-1][y-1] = dist[y-1][x-1] = 1

        return dist

    def floydWarshall(self, dist: List[List[int]], n: int) -> None:
        for k in range(n):
            for i in range(n):
                for j in range(n):
                    dist[i][j] = min(dist[i][j], dist[i][k] + dist[k][j])

    def countPairsForEachDistance(self, dist: List[List[int]], n: int) -> List[int]:
        result = [0] * n
        for i in range(n):
            for j in range(n):
                if i != j:
                    result[dist[i][j] - 1] += 1
        return result
```

## D.4 EFFICIENCY

**Good Case**: The model successfully generates code with a time complexity of $O(n)$ and a space complexity of $O(n)$. ($\text{NI}_T = 100.00, \text{NI}_S = 100.00$)

```
# Input
You are given a 0-indexed string s, a string a, a string b, and an integer k.
An index i is beautiful if:

0 <= i <= s.length - a.length
s[i..(i + a.length - 1)] == a
```

```
There exists an index j such that:

0 <= j <= s.length - b.length
s[j..(j + b.length - 1)] == b
|j - i| <= k

Return the array that contains beautiful indices in sorted order from smallest to largest.

Example 1:

Input: s = "isawsquirrelnearmysquirrelhouseohmy", a = "my", b = "squirrel", k = 15
Output: [16,33]
Explanation: There are 2 beautiful indices: [16,33].
- The index 16 is beautiful as s[16..17] == "my" and there exists an index 4 with s[4..11] ==
    "squirrel" and |16 - 4| <= 15.
- The index 33 is beautiful as s[33..34] == "my" and there exists an index 18 with s[18..25]
    == "squirrel" and |33 - 18| <= 15.
Thus we return [16,33] as the result.

Example 2:

Input: s = "abcd", a = "a", b = "a", k = 4
Output: [0]
Explanation: There is 1 beautiful index: [0].
- The index 0 is beautiful as s[0..0] == "a" and there exists an index 0 with s[0..0] == "a"
    and |0 - 0| <= 4.
Thus we return [0] as the result.

Constraints:

1 <= k <= s.length <= 5 * 105
1 <= a.length, b.length <= 5 * 105
s, a, and b contain only lowercase English letters.

Please complete the code below to solve above problem, and make sure that the time complexity
    is $O(n)$ and the space complexity is $O(n)$:
```python
class Solution:
    def beautifulIndices(self, s: str, a: str, b: str, k: int) -> List[int]:

```
```

```python
# Output
class Solution:
    def beautifulIndices(self, s: str, a: str, b: str, k: int) -> List[int]:
        def kmp_search(text, pattern):
            m = len(pattern)
            n = len(text)

            # Compute LPS array
            lps = [0] * m
            length = 0
            i = 1
            while i < m:
                if pattern[i] == pattern[length]:
                    length += 1
                    lps[i] = length
                    i += 1
                elif length != 0:
                    length = lps[length - 1]
                else:
                    lps[i] = 0
                    i += 1

            # KMP search
            result = []
            i = j = 0
            while i < n:
                if pattern[j] == text[i]:
                    i += 1
                    j += 1
                if j == m:
                    result.append(i - j)
                    j = lps[j - 1]
                elif i < n and pattern[j] != text[i]:
                    if j != 0:
```

```
                        j = lps[j - 1]
                else:
                    i += 1
        return result

    # Find all occurrences of 'a' and 'b' in 's'
    a_indices = kmp_search(s, a)
    b_indices = kmp_search(s, b)

    # Find beautiful indices
    beautiful_indices = []
    j = 0
    for i in a_indices:
        while j < len(b_indices) and b_indices[j] < i - k:
            j += 1
        if j < len(b_indices) and abs(b_indices[j] - i) <= k:
            beautiful_indices.append(i)

    return beautiful_indices
```

**Bad Case**: The model fails to generate code with a time complexity of $O(n)$. ($\mathrm{NI}_T = 0$)

```
# Input
You are given a 0-indexed array of positive integers nums.
In one operation, you can swap any two adjacent elements if they have the same number of set
    bits. You are allowed to do this operation any number of times (including zero).
Return true if you can sort the array, else return false.

Example 1:

Input: nums = [8,4,2,30,15]
Output: true
Explanation: Let's look at the binary representation of every element. The numbers 2, 4, and 8
     have one set bit each with binary representation "10", "100", and "1000" respectively.
     The numbers 15 and 30 have four set bits each with binary representation "1111" and
     "11110".
We can sort the array using 4 operations:
- Swap nums[0] with nums[1]. This operation is valid because 8 and 4 have one set bit each.
     The array becomes [4,8,2,30,15].
- Swap nums[1] with nums[2]. This operation is valid because 8 and 2 have one set bit each.
     The array becomes [4,2,8,30,15].
- Swap nums[0] with nums[1]. This operation is valid because 4 and 2 have one set bit each.
     The array becomes [2,4,8,30,15].
- Swap nums[3] with nums[4]. This operation is valid because 30 and 15 have four set bits each
    . The array becomes [2,4,8,15,30].
The array has become sorted, hence we return true.
Note that there may be other sequences of operations which also sort the array.

Example 2:

Input: nums = [1,2,3,4,5]
Output: true
Explanation: The array is already sorted, hence we return true.

Example 3:

Input: nums = [3,16,8,4,2]
Output: false
Explanation: It can be shown that it is not possible to sort the input array using any number
     of operations.

Constraints:

1 <= nums.length <= 100
1 <= nums[i] <= 28

Please complete the code below to solve above problem, and make sure that the time complexity
     is $O(n)$:
```python
class Solution:
    def canSortArray(self, nums: List[int]) -> bool:

```

```
# Output
class Solution:
    def canSortArray(self, nums: List[int]) -> bool:
```

```python
    def count_set_bits(n):
        return bin(n).count('1')

    n = len(nums)
    set_bits = [count_set_bits(num) for num in nums]

    # Group elements with the same number of set bits
    groups = []
    current_group = [nums[0]]
    current_bits = set_bits[0]

    for i in range(1, n):
        if set_bits[i] == current_bits:
            current_group.append(nums[i])
        else:
            groups.append(current_group)
            current_group = [nums[i]]
            current_bits = set_bits[i]

    groups.append(current_group)

    # Sort each group
    sorted_groups = [sorted(group) for group in groups]

    # Check if the sorted groups are in ascending order
    prev_max = 0
    for group in sorted_groups:
        if group[0] < prev_max:
            return False
        prev_max = group[-1]

    return True
```

