# OpenReview forum: "Beyond Correctness: Benchmarking Multi-dimensional Code Generation for Large Language Models"
_ICLR.cc/2025/Conference — ICLR 2025 Conference Withdrawn Submission_

### Official Review · Reviewer_uCC2 · 2024-11-03

**Soundness:** 2
**Presentation:** 3
**Contribution:** 1
**Rating:** 3
**Confidence:** 5

**Summary:**

This paper aims to develop code generation benchmark that focuses on objectives beyond correctness.
Specifically, this paper develops the RACE benchmark to cover readability, maintainability, correctness, and efficiency.
Additionally, this paper studies the performance of a comprehensive set of LLMs.

**Strengths:**

+ This work focuses on evaluating code generation of LLMs which is an important problem
+ Studywise this work evaluates a comprehensive set of LLMs and covers some recent LLMs

**Weaknesses:**

- There have been various published evaluation papers studying the aspect beyond code correctness [1,2,3]. For example, [1,2] covers the additional dimensions introduced in this paper, i.e., efficiency/maintainability/readability. Other non-correctness dimensions such as self-consistency and security have also been studied. Unfortunately, these prior papers seem to be ignored by this paper.
- There is little new technical design/novelty in the evaluation pipeline except for adding customized requirements.
- Overall this paper also did not bring new coding tasks but rather reused existing benchmarks.
- The metric design for readability and maintainability is debatable as these are subjective dimensions where the corresponding metrics should be prompt-defined.
- It seems the efficiency metric relies on execution runtime, whose value can be platform-dependent and impacted by system noises. Prior work [2,4] have proposed solution metrics to address this issue.
- This work did not talk about how to ablate correctness impact as different LLMs can have different sets of correct solutions.

[1] NoFunEval: Funny How Code LMs Falter on Requirements Beyond Functional Correctness. COLM 2024.
[2] Evaluating Language Models for Efficient Code Generation. COLM 2024.
[3] Beyond Accuracy: Evaluating Self-Consistency of Code Large Language Models with IdentityChain. ICLR 2024
[4] Learning Performance Improving Code Edits. ICLR 2024.

**Questions:**

- What's the novelty and contribution of this work considering the prior closely related work [1-4]
- When evaluating non-functional dimensions, how does this work deal with the associated correctness when computing the metrics

---

> ### Author Response · Authors · 2024-11-22
> **Rebuttal by Authors**
>
> Dear reviewer uCC2,
>
> Thank you for reviewing our manuscript and providing valuable feedback. We have carefully considered your comments and have prepared detailed responses to address each of your concerns.
>
> > W1 & W2 & W3 & Q1: What's the novelty and contribution of this work considering the prior closely related work?
>
> We would like to reiterate the contributions and value of our paper:
>
> - Based on realistic software engineering development scenarios, we propose a comprehensive evaluation framework for code generation that encompasses code correctness, readability, maintainability, and efficiency, addressing the gap between LLM-generated code and real-world scenarios, and steering code LLMs towards comprehensive development.
>
> - Based on the evaluation framework, we design representative customized requirements along with corresponding quantitative metrics, thereby constructing a comprehensive RACE benchmark. This allows for a convenient and reliable evaluation of the model's multidimensional code generation capabilities.
>
> - Using the RACE benchmark, we evaluate and analyze 28 LLMs, uncovering certain model deficiencies and providing valuable insights to guide the future development of code LLM. For instance, when the correctness-based benchmarks serve as guiding indicators, some LLMs excel in correctness but exhibit significant deficiencies in other dimensions, thereby failing to meet the demands of real-world development scenarios adequately. Furthermore, future code LLMs should enhance their ability to handle complex instructions through the design of novel learning algorithms, advancing toward practical software development agents.
>
> **In summary, we believe our paper underscores the importance of enhancing code LLMs' capability to generate high-quality code across multiple dimensions beyond correctness, revealing future directions for the development of code LLMs.**
>
> > W4: The metric design for readability and maintainability is debatable as these are subjective dimensions where the corresponding metrics should be prompt-defined.
>
> Readability and maintainability, due to their demand-dependent nature, are indeed subjective dimensions. However, our objective is to address the challenges associated with quantifying these demand-dependent characteristics. To this end, we propose a multidimensional quantitative evaluation framework and corresponding benchmarks. Through comprehensive evaluation and systematic analysis across 28 LLMs, we demonstrate that our benchmarks effectively assess the models' capabilities in meeting the requirements associated with the demand-dependent dimensions.
>
> > W5: It seems the efficiency metric relies on execution runtime, whose value can be platform-dependent and impacted by system noises. Prior work have proposed solution metrics to address this issue.
>
> We monitor the execution time and memory usage of the code within a Docker-based virtual environment, which is independent of the actual platform, thereby minimizing the impact of extraneous noise on the evaluation results. **As the focus of our paper is to provide valuable insights guiding the future development of LLMs**, the specific details of the assessment have not been included in the manuscript. We will add these details in the revision.
>
> > W6 & Q2: This work did not talk about how to ablate correctness impact as different LLMs can have different sets of correct solutions.
>
> We evaluate the correctness of the code based on the pass rate of test cases, which is independent of the different models having various correct solutions.
>
> **In conclusion, we hope that our responses provided address your primary concerns. If you have any additional questions or concerns, please do not hesitate to reach out.**

---

> > ### Author Response · Authors · 2024-11-26
> > **A kind reminder**
> >
> > Dear reviewer uCC2,
> >
> > We hope that our responses provided herein address your primary concerns. We want to gently remind you that the deadline for the discussion period is approaching. Please do not hesitate to inform us of any additional concerns or questions you may have.

---

> > ### Comment · Reviewer_uCC2 · 2024-11-26
> >
> > Thanks the authors for the reply. I prefer to maintain the current score given that the responses did not technically resolve my concerns.

---

### Official Review · Reviewer_sBVp · 2024-11-03

**Soundness:** 2
**Presentation:** 2
**Contribution:** 1
**Rating:** 3
**Confidence:** 5

**Summary:**

The paper proposes RACE, a new benchmark evaluating 4 dimensions of code generation: Readability, mAintainability, Correctness, and Efficiency. The authors claim that existing benchmarks primarily address code accuracy, neglecting the other aspects. For each dimension, the authors designed various types of user requirements to assess the model's ability to generate correct code that also meets user demands. A study of 28 LLMs using the RACE reveals that: 1) correctness-focused benchmarks are insufficient for real-world code requirements, while RACE offers a comprehensive evaluation; 2) RACE helps mitigate data contamination risks; 3) even advanced code LLMs struggle with complex, customized instructions; and 4) most LLMs show biases toward specific coding styles.

**Strengths:**

- In addition to the correctness, the paper evaluates multiple dimensions of code generation, including readability, maintainability, and efficiency, which most existing benchmarks are lack of. This is a good motivation.
- The provided artifact is well documented and seems to be reproducible. The authors also did a wide range of evaluations, involving 28 LLMs, open or closed, with different sizes.
- The authors showed that RACE is more robust to contaminations. It's also interesting that readability seems to be a critical
indicator of overall code quality.

**Weaknesses:**

1. The 4 dimensions chosen for constructing RACE are not convincing. While the authors claimed that the philosophy of the design comes from the demands for code quality in software engineering, it is unclear why these dimensions they are eventually selected.
2. The data source is quite limited, only covering 4 Python datasets. These datasets are more focused on self-contained code, limiting the scope of the paper. Also, if RACE is multi-dimensional, why not treat multi-language coding as another dimension?
3. The core contribution of the paper is weak. Actually, many high-quality correctness and code efficiency benchmarks exist in the wild. For correctness, besides HumanEval+, MBPP+, LeetCode, and ClassEval as mentioned in the paper, there are BigCodeBench, LiveCodeBench, MultiPL-E, etc. For efficiency, there are PIE and EvalPerf. Then the paper only contributed to readability and maintainability, which are much less important than correctness and efficiency.

**Questions:**

1. See the weaknesses mentioned above.
2. Can the authors justify the 4 selected dimensions? There are many other important aspects like security, documentation, modularity, robustness... What's the reason for not considering them?
3. How does RACE add value compared to various existing coding benchmarks, such as those discussed in "Weakness 3"?

---

> ### Author Response · Authors · 2024-11-22
> **Rebuttal by Authors (Part 1)**
>
> Dear reviewer sBVp,
>
> Thank you for reviewing our manuscript and providing valuable feedback. We have carefully considered your comments and have prepared detailed responses to address each of your concerns.
>
> > W1 & Q1: The 4 dimensions chosen for constructing RACE are not convincing.
> > Q2: Can the authors justify the 4 selected dimensions? There are many other important aspects like security, documentation, modularity, robustness...
>
> The 4 dimensions of correctness, readability, maintainability, and efficiency are among the most significant factors influencing code quality in real-world development scenarios. Specifically:
>
> - Code correctness is the core foundation for meeting user requirements.
>
> - Code readability is crucial for enhancing understandability and reducing the time required for code maintenance; it is considered one of the important factors affecting code quality by developers [1].
>
> - Code maintainability plays a vital role in the long-term health of software and the efficiency of development teams [2-4], involving principles such as the Single Responsibility Principle in software engineering to ensure the overall maintainability of projects.
>
> - Code efficiency is closely related to user experience and the efficiency of business processes.
>
> We recognize that numerous factors influence code quality in real-world development scenarios; however, other dimensions may not be applicable within our current evaluation framework. Specifically:
>
> - Regarding security, differing security assessment standards across systems introduce inconsistencies, and the security of code often depends on its execution environment, making it challenging to provide a definitive, quantifiable requirement that aligns with our evaluation needs.
>
> - Regarding documentation and modularity, our evaluation framework incorporates factors such as code comment and modularity.
>
> - Regarding robustness, it is not a direct factor affecting code quality and will not be addressed in our discussion.
>
> Consequently, we plan to expand our evaluation framework in future work to include additional important quality dimensions.
>
> > W2: The data source is quite limited, only covering 4 Python datasets. These datasets are more focused on self-contained code, limiting the scope of the paper.
>
> From the perspective of data scale, we include a total of 3,468 evaluation cases, which exceeds the scale of the vast majority of existing code benchmarks. From the perspective of data diversity, our data source encompasses coding exercises, coding competition questions, and class-level code generation tasks. Meanwhile, we have considered using repository-level benchmarks as our data source. However, after careful investigation, we found that existing repository-level benchmarks do not meet our evaluation requirements. Below are the detailed reasons:
>
> - For readability, we assess the ability of LLMs to generate code that functions correctly and meets user-specific needs. To the best of our knowledge, current repository-level benchmarks are in the form of code completion or summarization. This means that, given the related context, the task involves generating completed code or generating content to be edited based on specific editing requirements, leading LLMs to tend toward consistency with the style of the original code. **Therefore, these benchmarks are not suitable for assessing the model's ability to meet readability requirements.** Notably, RepoBench and CrossCodeEval focus on next-line prediction tasks, which are not suitable for assessing code readability. In contrast, we select widely used datasets that follow a problem-solution format, such as HumanEval+ and MBPP+, which are suitable for modifying entire solutions to meet specific readability requirements.
>
> - For maintainability, there are currently no code generation tasks for entire software systems targeted at LLMs. Moreover, **existing repository-level code generation tasks predominantly focus on local completion tasks—such as line-level, block-level, and function-level—which do not significantly impact the maintainability of an entire repository.** In contrast, we choose ClassEval featured by class-level code generation, which is more suitable for calculating the Maintainability Index to evaluate maintainability.
>
> - For efficiency, repository-level code generation, stemming from real development scenarios, often involves complex, cross-file context dependencies. As a result, **it is challenging to clearly define the varying efficiency requirements and standards,** making it difficult to meet our evaluation requirements regarding variances in efficiency. In contrast, LeetCode features complex algorithmic problems that can be addressed with solutions of varying complexity, making it more appropriate for evaluating code generation with specific complexity requirements.

---

> > ### Author Response · Authors · 2024-11-22
> > **Rebuttal by Authors (Part 2)**
> >
> > > W2: If RACE is multi-dimensional, why not treat multi-language coding as another dimension?
> >
> > While multi-programming languages represent an important dimension, our work focuses on exploring the multidimensional aspects of code quality, **where programming language is not considered a component of code quality.** Beyond our current scope, we aim to extend our future work into the realm of multi-programming languages, further investigating the performance and code preferences of code LLMs across different programming languages.
> >
> > > W3 & Q3: The core contribution of the paper is weak. How does RACE add value compared to various existing coding benchmarks?
> >
> > We would like to reiterate the contributions and value of our paper:
> >
> > - Based on realistic software engineering development scenarios, we propose a comprehensive evaluation framework for code generation that encompasses code correctness, readability, maintainability, and efficiency, addressing the gap between LLM-generated code and real-world scenarios, and steering code LLMs towards comprehensive development.
> >
> > - Based on the evaluation framework, we design representative customized requirements along with corresponding quantitative metrics, thereby constructing a comprehensive RACE benchmark. This allows for a convenient and reliable evaluation of the model's multidimensional code generation capabilities.
> >
> > - Using the RACE benchmark, we evaluate and analyze 28 LLMs, uncovering certain model deficiencies and providing valuable insights to guide the future development of code LLM. For instance, when the correctness-based benchmarks serve as guiding indicators, some LLMs excel in correctness but exhibit significant deficiencies in other dimensions, thereby failing to meet the demands of real-world development scenarios adequately. Furthermore, future code LLMs should enhance their ability to handle complex instructions through the design of novel learning algorithms, advancing toward practical software development agents.
> >
> > **In summary, we believe our paper underscores the importance of enhancing code LLMs' capability to generate high-quality code across multiple dimensions beyond correctness, revealing future directions for the development of code LLMs.**
> >
> > **In conclusion, we hope that our responses provided address your primary concerns. If you have any additional questions or concerns, please do not hesitate to reach out.**
> >
> > ### References
> >
> > [1] Börstler, Jürgen, et al. "Developers talking about code quality." Empirical Software Engineering 28.6 (2023): 128.
> >
> > [2] Heitlager, Ilja, Tobias Kuipers, and Joost Visser. "A practical model for measuring maintainability." 6th international conference on the quality of information and communications technology (QUATIC 2007). IEEE, 2007.
> >
> > [3] Coleman, Don, et al. "Using metrics to evaluate software system maintainability." Computer 27.8 (1994): 44-49.
> >
> > [4] Deissenboeck, Florian, et al. "An activity-based quality model for maintainability." 2007 IEEE International Conference on Software Maintenance. IEEE, 2007.

---

> > > ### Author Response · Authors · 2024-11-26
> > > **A kind reminder**
> > >
> > > Dear reviewer sBVp,
> > >
> > > We hope that our responses provided herein address your primary concerns. We want to gently remind you that the deadline for the discussion period is approaching. Please do not hesitate to inform us of any additional concerns or questions you may have.

---

> > > ### Comment · Reviewer_sBVp · 2024-11-26
> > >
> > > Thanks to the authors for the responses. However, the core contributions of the paper are still limited, which do not add enough extra values compared to existing benchmarks. I will keep my score.

---

### Official Review · Reviewer_JZ3b · 2024-11-04

**Soundness:** 2
**Presentation:** 2
**Contribution:** 1
**Rating:** 3
**Confidence:** 4

**Summary:**

Prior coding benchmarks primarily assess the accuracy of generated code - whether it follows natural language instructions and produces a correct answer. This paper introduces a new benchmark called RACE, which evaluates additional aspects beyond functionality, namely readability, maintainability, and efficiency. For example, each problem description adds additional requirements like “use camel/snake case for function names”, “each line is less than 60 characters long and each function is less than 20 lines long”, “add comments for each line in each function”, “use two additional sub-functions”, “use for/while”, “make sure that the time/space complexity is X”. Evaluation results show that most LLMs cannot follow such customized requirements, and the best model o1-mini has an overall score of only 63.5.

**Strengths:**

- The paper highlights an important problem. Readability, maintainability, and efficiency are important dimensions of code quality, yet understudied in prior benchmarks.

- The authors build their new benchmark RACE by adding additional requirements on existing, widely-adopted benchmarks (HumanEval+, MBPP+, ClassEval, LeetCode), and provide a comprehensive evaluation on 28 representative LLMs.

**Weaknesses:**

The motivation of this paper is somewhat unclear and unconvincing. In the introduction, the authors emphasize that code readability, maintainability, and efficiency are crucial properties for deciding code quality, and that existing benchmarks fail to distinguish between high-quality code from merely correct code. This is suggesting that RACE is designed to address this challenge by assessing these readability/maintainability/efficiency dimensions. However, RACE actually does not directly measure these qualities (except for the MI metric); rather, it measures the instruction-following ability, such as “use camel case for function names” or “use two additional sub-functions”. I agree with the authors that (1) readability/maintainability dimensions are hard to capture with a single metric like accuracy, and (2) some dimensions are demand-dependent. However, it seems to me that to tackle these challenges, more diverse metrics are needed to quantify and approximate different dimensions. For instance, tools like CodeQL can analyze source code for security issues, code smell [a] is often used for maintainability, and efficiency can be assessed with time efficiency and space efficiency metrics.

The authors seem to address a different problem - whether LLMs can follow coding style instructions. However, the styles studied are often superficial (e.g., naming conventions, lines of function, comment density). Consequently, a high RACE score might indicate that an LLM is better at understanding the ~7 types of code style instructions or at following specific coding styles, rather than reflecting actual code quality. For example, the model can easily boost its score by writing dummy code comments or dummy sub-functions.

Additionally, some test settings appear unnatural. Requiring the model to solve any problem using 1, 2, or 3 functions, for instance, may be impractical: some problems are too complex for a single function, while 3 functions may be too many for simple problems. Furthermore, as discussed in the paper, some LLMs may have an inherent preference bias towards specific styles (e.g., camel-case v.s. snake-case), and they cannot follow specific style-related instructions well, probably because of lack of instruction-tuning data in this domain. I wonder if better prompting, such as chain-of-thought or simply giving a few examples for camel-case naming, would significantly increase the IF accuracy of LLMs.

Human assessment is often essential for code quality studies, as it is very challenging to automatically evaluate source code quality. It would be beneficial to conduct a user study to verify whether RACE aligns with human preference, i.e., whether higher scores on RACE correlates with perceived better readability and quality for developers.

Lastly, I find the experimental setup of section 4.3 unconvincing. Using NL-to-code datasets like HumanEval+ as the contaminated dataset and using Magicoder-OSS-Instruct as the clean dataset may be problematic. Fine-tuning a model on such a small, domani-specific dataset (like HumanEval and MBPP) for 10 epochs would not only cause serious memorization issues, but also potentially impair the general instruction-following capabilities. This is because the instructions in HumanEval-like benchmarks usually only contain functionality requirements, and lack diversity (e.g., no coding style requirements). On the other hand, instruction-tuning datasets like Magicoder-OSS-Instruct include a wide range of instructions and potentially cover some scenarios similar to RACE. A more realistic contamination setup might involve mixing the correctness benchmark problems with instruction tuning datasets.


[a] Refactoring: Improving the Design of Existing Code.

**Questions:**

1. Could you please clarify the motivation of this paper?
2. Could you please justify the setup for section 4.3 and address my comments?

---

> ### Author Response · Authors · 2024-11-22
> **Rebuttal by Authors (Part 1)**
>
> Dear reviewer JZ3b,
>
> Thank you for reviewing our manuscript and providing valuable feedback. We have carefully considered your comments and have prepared detailed responses to address each of your concerns.
>
> > W1 & Q1: RACE actually does not directly measure these qualities; rather, it measures the instruction-following ability. It seems to me that to tackle these challenges, more diverse metrics are needed to quantify and approximate different dimensions.
>
> As mentioned in the Introduction Section of our paper, dimensions beyond correctness are demand-dependent. Accordingly, we set different important and representative user instruction requirements for these dimensions to evaluate the model's capability to generate code that is not only correct but also satisfies user needs across multiple dimensions. **The instruction-following ability of LLMs serves as the foundation for achieving this objective, which is also critical for the utilization of code LLMs in real-world development scenarios [1-2].**
>
> For the quantitative evaluation framework proposed in our paper:
>
> 1. **We analyze the most significant aspects influencing the quality of demand-dependent dimensions derived from software engineering development scenarios.** For instance, referencing relevant software engineering work [3], maintaining a consistent programming style is crucial for enhancing understandability and reducing maintenance time for code. Additionally, factors such as code length, naming conventions, and code comments contribute distinctly and importantly to code comprehensibility, leading us to select these aspects as part of the requirement set.
>
> 2. **We acknowledge that security issues and code smells are also significant dimensions and that we have previously considered incorporating them into our benchmark.** However, we find that these dimensions are not currently applicable to our evaluation framework. For instance, regarding code smells, guidelines for addressing dead code or data clumps tend to vary across different projects, making it challenging to establish a generalized quantitative metric, which does not align with our evaluation requirements. Meanwhile, we will consider expanding our evaluation framework in future work to include other significant dimensions, such as security issues and code smells.

---

> > ### Author Response · Authors · 2024-11-22
> > **Rebuttal by Authors (Part 2)**
> >
> > > W2: The authors seem to address a different problem - whether LLMs can follow coding style instructions.
> >
> > In the proposed quantitative evaluation framework, we not only incorporate requirements related to code style but also address aspects of code maintainability and efficiency. With respect to code maintainability, we include requirements for the model to generate code with varying levels of modularity. Regarding code efficiency, we require the model to produce different versions of code characterized by distinct time and space complexities.
> >
> > To address the issue that the RACE score does not accurately reflect true code quality and that models may easily cheat, we randomly sample 1,000 examples from all models involved in our paper. We evaluate the generated solutions using abstract syntax tree analysis to determine whether they contained identity functions (dummy functions) or empty comments (dummy comments). The experimental results indicated that no examples included identity functions and only 0.2% contained empty comments. Consequently, **we find that the current models almost do not exhibit such behavior.** Taking CodeLlama-7b-Instruct-hf, which has relatively weaker coding capabilities, as an example, the following are relevant examples of the generated results:
> >
> > 1. Code comment:
> >
> > ```
> > def count_distinct_chars(s):
> >     # Convert the string to lowercase
> >     s = s.lower()
> >     # Create a set to store the distinct characters
> >     distinct_chars = set()
> >     # Iterate over the characters in the string
> >     for c in s:
> >         # If the character is not in the set, add it
> >         if c not in distinct_chars:
> >             distinct_chars.add(c)
> >     # Return the length of the set, which is the number of distinct characters
> >     return len(distinct_chars)
> > ```
> >
> > 2. Modularity:
> >
> > ```
> > class Solution:
> >     def areSimilar(self, mat: List[List[int]], k: int) -> bool:
> >         def cyclicShift(row: List[int], shift: int) -> List[int]:
> >             return row[shift:] + row[:shift]
> >
> >         def isSimilar(row1: List[int], row2: List[int]) -> bool:
> >             return all(x == y for x, y in zip(row1, row2))
> >
> >         for i in range(k):
> >             oddRows = [cyclicShift(row, i) for i, row in enumerate(mat) if i % 2 == 1]
> >             evenRows = [cyclicShift(row, i) for i, row in enumerate(mat) if i % 2 == 0]
> >             if not all(isSimilar(row1, row2) for row1, row2 in zip(oddRows, evenRows)):
> >                 return False
> >
> >         return True
> > ```
> >
> > Thank you for your suggestion. **We develop the following automated processing methods to prevent instances of model cheating in the future:**
> >
> > 1. For code comments, we assess whether they contain substantive content. Comments that contain comment syntax but lack any meaningful content will be excluded.
> >
> > 2. For modularity, we check whether the generated sub-functions are identity functions; if they are, they will also be excluded.
> >
> > We will include the relevant analysis results and automated processing methods in the revision.
> >
> > > W3: Some test settings appear unnatural. Requiring the model to solve any problem using 1, 2, or 3 functions, for instance, may be impractical: some problems are too complex for a single function, while 3 functions may be too many for simple problems.
> >
> > **For modularity within maintainability, we establish the following guidelines for choosing evaluating data:** 1) the evaluation data should possess a certain level of complexity, enabling the design of reasonable and intricate user requirements; 2) the evaluation tasks need to be accurately assessable.
> >
> > Given that LeetCode data encompasses algorithmic problems with varying degrees of difficulty, and that these problems can be deterministically evaluated based on abstract syntax trees and the pass rates of test cases, we choose to use LeetCode data. Moreover, the algorithmic problems in LeetCode can typically be divided into several steps based on processing sequences and implementation logic, making them suitable for resolution using multiple functions. It is also inherently feasible to consolidate several steps within a single function, thereby allowing resolution with just one function as well.

---

> > > ### Author Response · Authors · 2024-11-22
> > > **Rebuttal by Authors (Part 3)**
> > >
> > > > W4: I find the experimental setup of section 4.3 unconvincing. A more realistic contamination setup might involve mixing the correctness benchmark problems with instruction tuning datasets.
> > >
> > > Our experimental setup for data leakage is convincing. The discrepancy in instruction-following capabilities between the contaminated and clean models does not stem from overfitting during the training on contaminated data, but rather from the models' failure to genuinely enhance their ability to generate code that meets user requirements. Specifically, in our experimental setup, we first trained the model on Code-Alpaca for 3 epochs to ensure that it possesses a certain level of instruction-following ability before the experimental evaluations. We believe this is analogous to the role of incorporating instruction data into the contamination set. Secondly, referencing related work on data leakage [4-7], we incorporated instruction-style benchmark data into the contamination set and included OOD instruction-following data in the clean set to ensure fairness in comparisons.
> > >
> > > Moreover, the performance discrepancy in instruction-following capability between the clean model and the contaminated model can primarily be attributed to data contamination leading the model to only learn to generate functionally correct code present in the training data. This results in increased accuracy on the benchmark without corresponding improvement on the RACE benchmark.
> > >
> > > To further address your concerns, we configure the contamination set to include benchmark data alongside a proportionate amount of general instruction-following data (Alpaca-GPT-4). The experimental results based directly on starcoderbase-7b are as follows:
> > >
> > > | Benchmark\Epochs      |   2   |         |        |   4   |         |        |   6   |         |        |   8   |         |        |   10  |         |        |
> > > |----------------|:-----:|:-------:|:------:|:-----:|:-------:|:------:|:-----:|:-------:|:------:|:-----:|:-------:|:------:|:-----:|:-------:|:------:|
> > > |                | Clean | w/ Test | $\Delta$ | Clean | w/ Test | $\Delta$ | Clean | w/ Test | $\Delta$ | Clean | w/ Test | $\Delta$ | Clean | w/ Test | $\Delta$ |
> > > | HumanEval+     | 32.3  |  29.9   |  -2.4  | 27.4  |  54.9   |  27.5  | 26.8  |  78.7   |  51.9  | 28.0  |  94.5   |  66.5  | 26.8  |  96.3   |  69.5  |
> > > | MBPP+          | 46.3  |  54.0   |  7.7   | 44.2  |  67.2   |  23.0  | 43.1  |  82.5   |  39.4  | 43.7  |  92.9   |  49.2  | 44.4  |  92.3   |  47.9  |
> > > | ClassEval      | 15.0  |  23.0   |  8.0   | 14.0  |  38.0   |  24.0  | 15.0  |  68.0   |  53.0  | 14.0  |  77.0   |  63.0  | 14.0  |  81.0   |  67.0  |
> > > | LeetCode       |  8.3  |   3.3   |  -5.0  |  5.0  |  20.6   |  15.6  |  7.8  |  51.1   |  43.3  |  6.1  |  87.2   |  81.1  |  5.6  |  94.4   |  88.8  |
> > > | RACE - Overall | 23.4  |  15.4   |  -8.0  | 20.6  |  26.1   |  5.5   | 21.4  |  34.4   |  13.0  | 20.6  |  46.4   |  25.8  | 20.7  |  48.3   |  27.6  |
> > > | RACE - IF rate | 58.0  |  40.1   | -17.9  | 56.4  |  47.4   |  -9.0  | 55.4  |  46.5   |  -8.9  | 54.9  |  46.7   |  -8.1  | 54.3  |  46.7   |  -7.7  |
> > >
> > > **We observe the same experimental conclusions as before, and the difference in instruction-following capability between the clean model and the contaminated model remains evident. Therefore, we assert that our experimental setup regarding data leakage is justified.**
> > >
> > > **In conclusion, we hope that our responses provided address your primary concerns. If you have any additional questions or concerns, please do not hesitate to reach out.**
> > >
> > > ### References
> > >
> > > [1] Hou, Xinyi, et al. "Large language models for software engineering: A systematic literature review." ACM Transactions on Software Engineering and Methodology (2023).
> > >
> > > [2] Jiang, Juyong, et al. "A Survey on Large Language Models for Code Generation." arXiv preprint arXiv:2406.00515 (2024).
> > >
> > > [3] Börstler, Jürgen, et al. "Developers talking about code quality." Empirical Software Engineering 28.6 (2023): 128.
> > >
> > > [4] Matton, Alexandre, et al. "On leakage of code generation evaluation datasets." arXiv preprint arXiv:2407.07565 (2024).
> > >
> > > [5] Ranaldi, Federico, et al. "Investigating the Impact of Data Contamination of Large Language Models in Text-to-SQL Translation." arXiv preprint arXiv:2402.08100 (2024).
> > >
> > > [6] Dong, Yihong, et al. "Generalization or memorization: Data contamination and trustworthy evaluation for large language models." arXiv preprint arXiv:2402.15938 (2024).
> > >
> > > [7] Ni, Shiwen, et al. "Training on the Benchmark Is Not All You Need." arXiv preprint arXiv:2409.01790 (2024).

---

> > > > ### Author Response · Authors · 2024-11-26
> > > > **A kind reminder**
> > > >
> > > > Dear reviewer JZ3b,
> > > >
> > > > We hope that our responses provided herein address your primary concerns. We want to gently remind you that the deadline for the discussion period is approaching. Please do not hesitate to inform us of any additional concerns or questions you may have.

---

> > > > > ### Comment · Reviewer_JZ3b · 2024-11-26
> > > > >
> > > > > Thank you to the authors for their response and additional experiments. However, my main concerns remain.
> > > > >
> > > > > **Q1 & W1 & W2**
> > > > >
> > > > > I fully agree that a consistent programming style and other readability/maintainability factors (e.g., code length, naming conventions, and code comments) are important. However, the proposed benchmark does not evaluate the “consistency” of coding styles, but instead test if the model can generate isolated code snippets that can follow certain code style requirements. My concern is that the model might struggle to understand certain code style requirements described in short natural language instruction, as chosen by the authors. However, the model could still follow a consistent coding style when provided with repository context. Consider the following alternative evaluation settings:
> > > > >
> > > > > (1) Repository context: For example, when given with context code that uses a specific naming convention (e.g., camel-case), the model might naturally adopt the same style. This setting seems more realistic than instructing the model directly to “use camel-case”.
> > > > >
> > > > > (2) Few-shot examples: Instructions like “add comments for each line in each function” comment can be ambiguous and/or unnatural. It is generally uncommon to add a comment for every single line. Instead, “line-by-line comments” usually refers to a verbose style that annotates each line when necessary, as opposed to function-level comments. As noted in my original review, providing few-shot examples in the prompt could potentially significantly improve the IF accuracy.
> > > > >
> > > > > Additionally, in my opinion, the “bad case” for line-by-line comment in the response to Reviewer SfYX is actually acceptable: The granularity is close to line-by-line in that it explains the functionality of each line while combining some lines when they work toward an atomic functionality (e.g., the if-else statement). This highlights my main concern: the low IF accuracy seems more related to instruction ambiguity or unnatural instructions, rather than the model’s inability to generate high-quality code when provided with clear and well-presented instructions.
> > > > >
> > > > > Finally, given the benchmark tested only a limited number of styles with fixed prompt, the evaluation seems biased.
> > > > >
> > > > > > It is also inherently feasible to consolidate several steps within a single function, thereby allowing resolution with just one function as well
> > > > >
> > > > > While I agree that, theoretically, any algorithm can be implemented within a single function, I question whether this is a practical or meaningful evaluation task. In my experience, some LeetCode tasks are better implemented using multiple functions. Some manual case studies could help justify the selection criteria and evaluation practicality.
> > > > >
> > > > > **W4**
> > > > >
> > > > > Thank you for the additional experiments. However, this does not fully address my concern.
> > > > >
> > > > > The main issue is that the instruction-style HumanEval/MBPP/... data contain only functionality requirements, making it OOD with respect to RACE requirements (even though it is in distribution when considering only functionality). This is not a fair comparison to the code instruction-following data (magicoder), which contain diverse code-related instructions and therefore more in-distribution with respect to RACE’s additional requirements.
> > > > >
> > > > > I am aware that the model is trained on Code-Alpaca, therefore it possesses a certain level of instruction-following ability before contamination fine-tuning. My concern is that fine-tuning on the small, close-domain contamination dataset for some epochs could lead to catastrophic forgetting, reducing the model’s ability to follow diverse instructions like the additional requirements in RACE. The new experiments mixing general instruction-following data with contamination benchmark data show similar trends, which partially addresses this concern. However, further details and evaluations are needed to further demonstrate a realistic contamination scenario. Specifically:  Does the fine-tuned model still retain its instruction-following ability (i.e.,  coding/general benchmarks outside the RACE benchmark) after fine-tuning the contamination dataset or clean dataset?

---

### Official Review · Reviewer_SfYX · 2024-11-04

**Soundness:** 3
**Presentation:** 3
**Contribution:** 3
**Rating:** 6
**Confidence:** 3

**Summary:**

The paper introduces the RACE benchmark for evaluating code generation by large language models (LLMs) across four dimensions: Readability, Maintainability, Correctness, and Efficiency.
By mimicking real-world software quality requirements, the benchmark is designed to address the limitations of current existing, correctness-focused benchmarks.
For each dimension of evaluation, the authors provide practical metrics with specific user demands.
Via empirical evaluation of 28 representative LLMs using RACE, the authors provide insights into code generation capabilities and validate the robustness of multidimensional evaluation as an effective tool in resisting the risk of data contamination, identifying areas where LLMs struggle beyond correctness, e.g., coding style and complex instructions.

**Strengths:**

* The RACE benchmark introduces a novel approach to code quality evaluation, reflecting real-world demands more closely (e.g., including user-specific requirements within each dimension) and advocates for more robust, practical benchmarks
* The multidimensional evaluation is sound, with well-defined metrics for each dimension, e.g., the use of established software quality metrics like the Maintainability Index is well-justified.
* The paper conducts an extensive evaluation across a wide range of models, including both open-source and closed-source models, to provide a comprehensive view of current LLM capabilities and limitations.
* The results support the main claims well, showing that models generally underperform in dimensions beyond correctness. The insight and highlight of several critical issues are useful, such as LLMs' biases toward certain coding styles, and performance degradation under complex requirements.
* The paper is well presented with a reasonable level of technical detail for each metric.

**Weaknesses:**

- The claim that RACE mitigates data contamination is somewhat speculative without further investigation or deeper examination: While RACE may lessen overfitting by constraining diverse, customized requirements, the inherent data contamination issues with large code datasets remain complex.
- While the benchmark is thorough, the study lacks concrete examples and case studies to compare failed or succeeded in each dimension
- The paper focuses on code generation models using Python-specific tasks. Expanding RACE to other languages (e.g., JavaScript, Java) would allow the benchmark to better generalize and expand the applicability.

**Questions:**

1. Regarding this statement "If these correctness-based benchmarks serve as guiding indicators and correctness alone becomes the sole criterion for driving LLM development, these models might end up memorizing the exact solutions from the training data instead of understanding the underlying principles or patterns. This overfitting implies the model may reproduce code that is highly similar to the training data during inference, leading to data leakage."? (Line 73), can you elaborate on why focusing on correctness leads to overfitting? and data leakage? assuming you provide large enough data for training and evaluation?
Any reference for these statements?

2. For the metrics:
- Could the authors provide a rationale for using MI as the primary metric for maintainability, given its known limitations? Would combining MI with additional maintainability metrics (e.g., code cohesion) offer a more comprehensive assessment?
- For the Normalized Index (NI) and proposed metrics, have you unit tested its consistency with provided complexity (or ground truth metrics)? And  How sensitive are the Normalized Indices (NIT, NIS) to minor improvements in time and space efficiency?

3. For the decline in performance with complex instructions, an alternative hypothesis could be that models lack generalization to uncommon or contradictory requirements, rather than simply not understanding multiple demands. Have the authors tested model tuning with various prompt instructions that might mitigate this issue?

4. For an alternative hypothesis, have you considered if the data contamination robustness observed could also stem from inherent limitations in LLMs’ instruction-following capabilities? This could validate whether the robustness is an incidental outcome or  a direct effect of the multidimensional approach.

---

> ### Author Response · Authors · 2024-11-22
> **Rebuttal by Authors (Part 1)**
>
> Dear reviewer SfYX,
>
> Thank you for reviewing our manuscript and providing valuable feedback. We have carefully considered your comments and have prepared detailed responses to address each of your concerns.
>
> > W1: The claim that RACE mitigates data contamination is somewhat speculative without further investigation or deeper examination.
>
> Due to the large scale and opaqueness of training data for current LLMs, it is almost impractical to completely eliminate dataset contamination within benchmarks. **Therefore, the value of RACE lies in mitigating the impact of data contamination on evaluation reliability, rather than in eliminating data contamination present in current benchmarks.** This is equally important for guiding the future development direction of code LLMs. Specifically, as demonstrated in the experiments on data contamination presented in Section 4.3 of the paper (see Table 3), data contamination resulted in the model learning to generate only the functionally correct code that appeared in the training data, without genuinely enhancing its capacity to generate code that meets user requirements. Consequently, there was no performance improvement on the RACE benchmark. Thus, RACE can serve as an effective tool for resisting data contamination.
>
> > W2: While the benchmark is thorough, the study lacks concrete examples and case studies to compare failed or succeeded in each dimension
>
> In the revision, we have included case studies corresponding to the four dimensions of RACE. Here are some representative cases:
>
> 1. Good Case from Claude-3.5-Sonnet: The model successfully generates code accompanied by line-by-line comments.
>
> ```
> def are_numbers_close(numbers, threshold):
>     # Sort the list of numbers in ascending order
>     sorted_numbers = sorted(numbers)
>
>     # Iterate through the sorted list, comparing adjacent pairs
>     for i in range(len(sorted_numbers) - 1):
>         # Calculate the difference between adjacent numbers
>         difference = sorted_numbers[i+1] - sorted_numbers[i]
>
>         # Check if the difference is less than the threshold
>         if difference < threshold:
>             # If a pair is found closer than the threshold, return True
>             return True
>
>     # If no pair is found closer than the threshold, return False
>     return False
> ```
>
> 2. Bad Case from Claude-3.5-Sonnet: The model fails to generate code accompanied by line-by-line comments.
>
> ```
> def binary_xor(a: str, b: str) -> str:
>     # Function to perform binary XOR on two input strings
>
>     # Ensure both input strings have the same length by padding with zeros
>     max_length = max(len(a), len(b))
>     a = a.zfill(max_length)
>     b = b.zfill(max_length)
>
>     # Initialize an empty result string
>     result = ""
>
>     # Iterate through each position in the input strings
>     for i in range(max_length):
>         # Perform XOR on corresponding bits and add to result
>         if a[i] == b[i]:
>             result += "0"
>         else:
>             result += "1"
>
>     # Remove leading zeros from the result
>     result = result.lstrip("0")
>
>     # Return "0" if result is empty (all zeros), otherwise return result
>     return result if result else "0"
> ```
>
> > W3: Expanding RACE to other languages would allow the benchmark to better generalize and expand the applicability.
>
> Expanding the code generation evaluation task to encompass additional programming languages is an essential development direction. Given that the metric calculations for RACE rely solely on static code analysis and runtime monitoring methods, efficient language migration can be achieved using existing tools such as SonarQube and New Relic.
>
> Due to our focus on the evaluation of multi-dimensional code generation capabilities, the assessment of code across multiple programming languages is beyond the current scope of our study. Therefore, we plan to extend our work into the multi-programming language domain in future research, further investigating the performance of code LLMs across various dimensions and preferences in different programming languages.

---

> > ### Author Response · Authors · 2024-11-22
> > **Rebuttal by Authors (Part 2)**
> >
> > > Q1: Can you elaborate on why focusing on correctness leads to overfitting? and data leakage? assuming you provide large enough data for training and evaluation? Any reference for these statements?
> >
> > If existing code benchmarks use correctness as the sole criterion for evaluation, this singular focus would influence the choices regarding the construction of training data and the design of training strategies for LLMs, leading to a prioritization of correctness as the only optimization target in both phases. Consequently, the model could primarily end up memorizing the functionally correct answers that appear in the training data, rather than developing the ability to generate solutions with broader applicability. This memorization strategy not only diminishes the model's generalization capability but also introduces potential data leakage, thereby reducing the model's capacity to effectively respond to novel situations. The decline in the true capabilities of the model due to data leakage has also been observed in numerous previous studies [1-3].
> >
> > > Q2: For the metric about MI, could the authors provide a rationale for using MI as the primary metric for maintainability, given its known limitations?
> >
> > In fact, in our earlier work, we utilized cohesion as a supplementary metric, but we found considerable overlap with the calculation method for MI. **To maximize the efficiency of our maintainability metrics assessment, we have retained only the MI metric in this version.** In the future, we will continue to explore maintainability metrics that align more closely with human judgment standards.
> >
> > > Q3: For the decline in performance with complex instructions, an alternative hypothesis could be that models lack generalization to uncommon or contradictory requirements, rather than simply not understanding multiple demands.
> >
> > First, **we do not provide a set of conflicting or uncommon demands**; rather, we select from a combination of naming conventions, length, annotations, modularity, and efficiency, wherein multiple constructed requirements can be satisfied simultaneously.
> >
> > Secondly, **the models we evaluate include several powerful instruction-type models that have been trained on large amounts of complex instruction data.** For instance, Section 4.2.2 of the Llama-3 technical report [4] and Section 3.5.1 of the DeepSeek-Coder-V2 technical report [5] illustrate the diversity of instruction data sources. Consequently, the models have encountered a sufficiently rich variety of instructions.
> >
> > We appreciate your suggestion and will consider exploring the impact of different training data and strategies on the models’ capacity for following complex instructions in the future.
> >
> > > Q4: Have you considered if the data contamination robustness observed could also stem from inherent limitations in LLMs’ instruction-following capabilities?
> >
> > As you mentioned, it is indeed possible that the observed robustness to data contamination is due to the model's insufficient instruction-following capability. However, it is precisely because the model merely memorized the functionally correct code from the training data during the contamination process, without genuinely enhancing its ability to generate code that meets user requirements. Consequently, the experimental results show an improvement in performance on the original data, while the performance on RACE remains almost unchanged.
> >
> > **In conclusion, we hope that our responses provided address your primary concerns. If you have any additional questions or concerns, please do not hesitate to reach out.**
> >
> > ### References
> >
> > [1] Matton, Alexandre, et al. "On leakage of code generation evaluation datasets." arXiv preprint arXiv:2407.07565 (2024).
> >
> > [2] Ranaldi, Federico, et al. "Investigating the Impact of Data Contamination of Large Language Models in Text-to-SQL Translation." arXiv preprint arXiv:2402.08100 (2024).
> >
> > [3] Dong, Yihong, et al. "Generalization or memorization: Data contamination and trustworthy evaluation for large language models." arXiv preprint arXiv:2402.15938 (2024).
> >
> > [4] Dubey, Abhimanyu, et al. "The Llama 3 Herd of Models." arXiv preprint arXiv:2407.21783 (2024).
> >
> > [5] Zhu, Qihao, et al. "DeepSeek-Coder-V2: Breaking the Barrier of Closed-Source Models in Code Intelligence." arXiv preprint arXiv:2406.11931 (2024).

---

> > > ### Comment · Reviewer_SfYX · 2024-11-26
> > > **Re: Official Comment by Authors**
> > >
> > > Thank you for your detailed responses and revision.

---

### Official Review · Reviewer_aRwS · 2024-11-04

**Soundness:** 2
**Presentation:** 3
**Contribution:** 2
**Rating:** 3
**Confidence:** 5

**Summary:**

The paper introduces the RACE benchmark, a novel multi-dimensional evaluation framework for assessing the quality of code generated by large language models (LLMs). Unlike traditional benchmarks that focus solely on code correctness, RACE evaluates LLMs across four dimensions: Readability, mAintainability, Correctness, and Efficiency. The authors develop specific metrics and customized requirements for each dimension, conducting a comprehensive evaluation of 28 representative LLMs. Their findings reveal significant limitations in current correctness-centric benchmarks and highlight the necessity for a holistic evaluation approach to guide LLM development towards generating high-quality, user-compliant code in real-world scenarios.

**Strengths:**

The paper addresses a critical gap in LLM evaluation by moving beyond correctness to include essential dimensions like readability, maintainability, and efficiency. The RACE benchmark is a good contribution, offering a comprehensive, demand-dependent evaluation framework that aligns closely with real-world software development needs. The empirical evaluation of a wide range of LLMs provides valuable insights into the capabilities and limitations of existing models. The paper’s methodology, which includes both static analysis and runtime monitoring, ensures robust and objective assessment, enhancing the credibility and applicability of its findings.

**Weaknesses:**

Although the benchmark addresses critical dimensions, the paper falls short in validating its framework through real-world case studies or industry-specific applications. Real-world validation would significantly enhance the credibility and practical relevance of the RACE benchmark. Incorporating case studies or pilot projects in industry settings could provide deeper insights into its applicability and effectiveness across diverse environments. Specifically, the paper should focus on the repository-level evaluation, as generating and executing code at this level offers a more comprehensive and practical measure of dimensions like maintainability and efficiency. For instance, the use of the Maintainability Index (MI), as defined by Coleman et al., is somewhat misplaced in this context. MI was designed for evaluating entire software systems rather than simple coding tasks typical in competitive coding benchmarks.

Regarding code efficiency, the motivation behind the Normalized Index (NI) metric is insufficiently explained. The authors should refer to established methodologies, such as those in [5][6], to justify the metric and outline a robust data collection pipeline with appropriate metrics for assessing code efficiency. This would provide a more grounded and insightful evaluation framework, ensuring that the NI metric accurately reflects real-world efficiency demands.


[1] RepoBench: Benchmarking Repository-Level Code Auto-Completion Systems, https://arxiv.org/abs/2306.03091

[2] DevEval: A Manually-Annotated Code Generation Benchmark Aligned with Real-World Code Repositories, https://arxiv.org/abs/2405.19856

[3] CrossCodeEval: A Diverse and Multilingual Benchmark for Cross-File Code Completion, https://arxiv.org/abs/2310.11248

[4] On the Impacts of Contexts on Repository-Level Code Generation, https://arxiv.org/abs/2406.11927v3

[5] Learning Performance-Improving Code Edits, https://arxiv.org/abs/2302.07867

[6] PerfCurator: Curating a large-scale dataset of performance bug-related commits from public repositories, https://arxiv.org/abs/2406.11731

**Questions:**

1) Have you considered validating the RACE benchmark through real-world case studies or pilot projects in industry settings? If so, what industries or types of projects do you envision as ideal test beds?

2) Given that the Maintainability Index (MI) was originally designed for entire software systems, how do you justify its application to individual code snippets or competitive coding benchmarks? Do you plan to develop or adapt other metrics that might be more appropriate for this scope?

3) The results indicate that LLMs struggle with complex, multi-faceted instructions. Have you explored specific training techniques or architectural adjustments that could improve these capabilities? What future directions do you see for enhancing LLMs’ instruction-following skills?

4) Since the evaluation depends heavily on user-defined requirements, what strategies do you recommend for minimizing bias or ensuring consistency in requirement selection? Would a standardized set of requirements or guidelines be feasible?

---

> ### Author Response · Authors · 2024-11-22
> **Rebuttal by Authors (Part 1)**
>
> Dear reviewer aRwS,
>
> Thank you for reviewing our manuscript and providing valuable feedback. We have carefully considered your comments and have prepared detailed responses to address each of your concerns.
>
> > W1 & Q1: Although the benchmark addresses critical dimensions, the paper falls short in validating its framework through real-world case studies or industry-specific applications.
>
> Initially, we considered using repository-level benchmarks as our data source. However, after careful investigation, we found that existing repository-level benchmarks do not meet our evaluation requirements. Therefore, in the current version of our benchmark, we choose coding exercises, coding competition problems, and class-level code generation tasks. Below are the detailed reasons:
>
> - For readability, we assess the ability of LLMs to generate code that functions correctly and meets user-specific needs. To the best of our knowledge, current repository-level benchmarks [1-5] are in the form of code completion or summarization. This means that, given the related context, the task involves generating completed code or generating content to be edited based on specific editing requirements, leading LLMs to tend toward consistency with the style of the original code. **Therefore, these benchmarks are not suitable for assessing the model's ability to meet readability requirements.** Notably, RepoBench [6] and CrossCodeEval [7] focus on next-line prediction tasks, which are not suitable for assessing code readability. In contrast, we select widely used datasets that follow a problem-solution format, such as HumanEval+ and MBPP+, which are suitable for modifying entire solutions to meet specific readability requirements.
>
> - For maintainability, there are currently no code generation tasks for entire software systems targeted at LLMs. Moreover, **existing repository-level code generation tasks predominantly focus on local completion tasks—such as line-level, block-level, and function-level—which do not significantly impact the maintainability of an entire repository.** In contrast, we choose ClassEval featured by class-level code generation, which is more suitable for calculating the Maintainability Index to evaluate maintainability.
>
> - For efficiency, repository-level code generation, stemming from real development scenarios, often involves complex, cross-file context dependencies. As a result, **it is challenging to clearly define the varying efficiency requirements and standards**, making it difficult to meet our evaluation requirements regarding variances in efficiency. In contrast, LeetCode features complex algorithmic problems that can be addressed with solutions of varying complexity, making it more appropriate for evaluating code generation with specific complexity requirements.
>
> To further address your concern, we sample 100 pieces of data from DevEval [8] to supplement our readability experiments, with the following results:
>
> | Model                |   Correctness  | Length | Comment |
> |----------------------|:--------------:|:------:|:-------:|
> | GPT-4o-mini          |        54.0    |  8.3   |  48.0   |
> | DeepSeek-V2.5        |      59.0      |  6.7   |  47.5   |
> | Llama-3-70B-Instruct |      57.0      |  4.0   |  42.0   |
>
> **We find consistent conclusions from the results of DevEval**, and we will include the complete experimental results in the revision.
>
>
> > W2: Regarding code efficiency, the motivation behind the Normalized Index (NI) metric is insufficiently explained.
>
> The two referenced works mentioned above and our NI metric are both based on comparisons of execution time or space utilization between the evaluated code and the ground-truth code. Furthermore, compared to these reference works, **the NI metric is capable of assessing customized requirements tailored to varying time and space complexities, thus aligning more closely with our evaluation needs.**
>
> For the collection of data on code efficiency, we employ the standardized pipeline:
>
> - Select problems from LeetCode that provide at least two official solutions with differing complexities.
>
> - For each selected problem, construct 100 random test cases and 1 edge case test case based on the problem's requirements.
>
> - In our Docker-based virtual runtime environment, validate the correctness of all solutions according to the pass rate of the test cases. Subsequently, monitor the execution time and memory consumption of solutions with differing complexities. If the execution time or memory usage of solutions with different complexities varies by more than 50%, the problem is included in the dataset for code efficiency evaluation.
>
> **Thus, using the standardized data collection pipeline, our proposed NI metric can accurately reflect the degree to which user-defined complexity requirements are satisfied.**

---

> > ### Author Response · Authors · 2024-11-22
> > **Rebuttal by Authors (Part 2)**
> >
> > > Q2: Given that the Maintainability Index (MI) was originally designed for entire software systems, how do you justify its application to individual code snippets or competitive coding benchmarks?
> >
> > As you mentioned, MI should be used to assess the entire software system; however, there is currently no code generation task that encompasses the whole system. Additionally, the existing repository-level code generation tasks focus on localized generation and do not significantly impact the overall maintainability of the repository. In contrast, ClassEval is designed for code generation at the class level, making it more suitable for evaluating maintainability through MI (see the response to W1).
> >
> > Meanwhile, in our earlier work, we utilized cohesion as a supplementary metric, but we found considerable overlap with the calculation method for MI. **To maximize the efficiency of our maintainability metrics assessment, we have retained only the MI metric in this version.** In the future, we will continue to explore maintainability metrics that align more closely with human judgment standards.
> >
> > > Q3: Have you explored specific training techniques or architectural adjustments that could improve these capabilities?
> >
> > Your proposed research direction of enhancing the model's ability to process complex instructions using specific training methods or architectures is highly valuable. However, Our paper focuses on the multi-dimensional evaluation of code generation quality. Consequently, we have not conducted related experimental analyses, although we do have some preliminary reflections. For instance, we believe that the o1-type reasoning approach represents a feasible path for LLMs to address complex instructions. By explicitly constructing execution pathways, LLMs can engage in a step-by-step process of thinking, generating, evaluating, and reflecting, thereby satisfying all fine-grained requirements embedded in complex instructions.
> >
> > Since methods for enhancing model capabilities across different dimensions fall beyond the scope of our current work, we leave the exploration of this reasoning approach for future work.

---

> > > ### Author Response · Authors · 2024-11-22
> > > **Rebuttal by Authors (Part 3)**
> > >
> > > > Q4: What strategies do you recommend for minimizing bias or ensuring consistency in requirement selection?
> > >
> > > As discussed in Section 3 of our paper, we adhere to the following standards for selecting requirements:
> > >
> > > 1. Readability: We draw upon existing software engineering literature on readability [9], emphasizing that maintaining a consistent programming style is crucial for enhancing comprehensibility and reducing code maintenance time. Additionally, excessively long lines of code may lead to screen truncation. A uniform naming convention assists developers in quickly understanding interface functionality, while code comments facilitate rapid comprehension of code logic. These factors each contribute significantly and distinctly to enhancing code comprehension, thus we select them as part of our requirement set.
> > >
> > > 2. Maintainability: We reference several quality models and the Single Responsibility Principle [10-12], thereby adopting the Maintainability Index and modularity as our corresponding requirements.
> > >
> > > 3. Efficiency: Based on widely accepted methods for evaluating code efficiency, we choose time complexity and space complexity as our relevant requirements.
> > >
> > > **In conclusion, we hope that our responses provided address your primary concerns. If you have any additional questions or concerns, please do not hesitate to reach out.**
> > >
> > > ### References
> > >
> > > [1] Zhang, Fengji, et al. "RepoCoder: Repository-Level Code Completion Through Iterative Retrieval and Generation." Proceedings of the 2023 Conference on Empirical Methods in Natural Language Processing. 2023.
> > >
> > > [2] Liu, Tianyang, Canwen Xu, and Julian McAuley. "RepoBench: Benchmarking Repository-Level Code Auto-Completion Systems." The Twelfth International Conference on Learning Representations.
> > >
> > > [3] Jimenez, Carlos E., et al. "SWE-bench: Can Language Models Resolve Real-world Github Issues?." The Twelfth International Conference on Learning Representations.
> > >
> > > [4] Wu, Di, et al. "Repoformer: Selective Retrieval for Repository-Level Code Completion." Forty-first International Conference on Machine Learning.
> > >
> > > [5] Hai, Nam Le, Dung Manh Nguyen, and Nghi DQ Bui. "REPOEXEC: Evaluate Code Generation with a Repository-Level Executable Benchmark." arXiv preprint arXiv:2406.11927 (2024).
> > >
> > > [6] Liu, Tianyang, Canwen Xu, and Julian McAuley. "Repobench: Benchmarking repository-level code auto-completion systems." arXiv preprint arXiv:2306.03091 (2023).
> > >
> > > [7] Ding, Yangruibo, et al. "Crosscodeeval: A diverse and multilingual benchmark for cross-file code completion." Advances in Neural Information Processing Systems 36 (2024).
> > >
> > > [8] Li, Jia, et al. "DevEval: A Manually-Annotated Code Generation Benchmark Aligned with Real-World Code Repositories." arXiv preprint arXiv:2405.19856 (2024).
> > >
> > > [9] Börstler, Jürgen, et al. "Developers talking about code quality." Empirical Software Engineering 28.6 (2023): 128.
> > >
> > > [10] Heitlager, Ilja, Tobias Kuipers, and Joost Visser. "A practical model for measuring maintainability." 6th international conference on the quality of information and communications technology (QUATIC 2007). IEEE, 2007.
> > >
> > > [11] Coleman, Don, et al. "Using metrics to evaluate software system maintainability." Computer 27.8 (1994): 44-49.
> > >
> > > [12] Deissenboeck, Florian, et al. "An activity-based quality model for maintainability." 2007 IEEE International Conference on Software Maintenance. IEEE, 2007.

---

> > > > ### Author Response · Authors · 2024-11-26
> > > > **A kind reminder**
> > > >
> > > > Dear reviewer aRwS,
> > > >
> > > > We hope that our responses provided herein address your primary concerns. We want to gently remind you that the deadline for the discussion period is approaching. Please do not hesitate to inform us of any additional concerns or questions you may have.

---

> > > > > ### Comment · Reviewer_aRwS · 2024-11-26
> > > > >
> > > > > Thank you for the responses. I believe the paper requires further revisions before it can be considered for acceptance, particularly concerning the assessment of maintainability metrics and their definition.
> > > > >
> > > > > Regarding the statement: **For maintainability, there are currently no code generation tasks for entire software systems targeted at LLMs** — I interpret this as highlighting a significant challenge. However, the authors should invest more effort in rethinking this metric and defining clearer dimensions for this benchmark.
> > > > >
> > > > > At this stage, I will retain my current score for the paper.

---

### Note · Authors · 2024-12-04

I have read and agree with the venue's withdrawal policy on behalf of myself and my co-authors.